# Impaired liver regeneration in aged mice can be rescued by silencing Hippo core kinases MST1 and MST2

Giulio Loforese[1], Thomas Malinka[1], Adrian Keogh[1], Felix Baier[1], Cedric Simillion[2], Matteo Montani[3], Thanos D Halazonetis[4] iD, Daniel Candinas[1] & Deborah Stroka[1],* iD

## Abstract

The liver has an intrinsic capacity to regenerate in response to injury or surgical resection. Nevertheless, circumstances in which hepatocytes are unresponsive to proliferative signals result in impaired regeneration and hepatic failure. As the Hippo pathway has a canonical role in the maintenance of liver size, we investigated whether it could serve as a therapeutic target to support regeneration. Using a standard two-thirds partial hepatectomy (PH) model in young and aged mice, we demonstrate that the Hippo pathway is modulated across the phases of liver regeneration. The activity of the core kinases MST1 and LATS1 increased during the early hypertrophic phase and returned to steady state levels in the proliferative phase, coinciding with activation of YAP1 target genes and hepatocyte proliferation. Moreover, following PH in aged mice, we demonstrate that Hippo signaling is anomalous in non-regenerating livers. We provide pre-clinical evidence that silencing the Hippo core kinases MST1 and MST2 with siRNA provokes hepatocyte proliferation in quiescent livers and rescues liver regeneration in aged mice following PH. Our data suggest that targeting the Hippo core kinases MST1/2 has therapeutic potential to improve regeneration in non-regenerative disorders.

**Keywords** aged liver; Hippo pathway; liver regeneration; MST; RNAi

**Subject Categories** Digestive System; Regenerative Medicine

## Introduction

Due to its intrinsic capacity to regenerate—the liver has been at the forefront of clinical regenerative medicine and enables surgical treatments for numerous hepatic disorders. Livers of patients with cancer, parasitic infections or diseases that result in acute or chronic liver failure can be resected or transplanted using living-related or split-liver donor organs. In each circumstance, patient recovery depends on the remnant liver mass to undergo compensatory growth to return to its original weight and functional capacity. Compensatory growth, or liver regeneration, following partial hepatectomy is a rapid and efficient process in which the normally quiescent hepatocyte population re-enters cell cycle and proliferates until the restoration of lost hepatic mass (Michalopoulos & DeFrances, 1997; Taub, 2004). However, there are cases in which regeneration fails and this results in hepatic insufficiency and acute hepatic failure, known clinically as small for size syndrome (SFSS) (Troisi et al, 2003; Zhong et al, 2006). Susceptible livers for SFSS include those with cirrhosis or hepatosteatosis. Interestingly, ageing is also an adverse factor for liver regeneration; several studies in mice and humans demonstrated an age-dependent decline in the liver's regenerative capacity following PH (Fry et al, 1984; Iakova et al, 2003; Ledda-Columbano et al, 2004; Schmucker & Sanchez, 2011). Although liver regeneration has been extensively investigated, many regulatory pathways remain elusive and to date, there is no non-invasive pharmacological means to support regeneration for clinical use. In this study, we investigated a potential targeted treatment to improve regeneration through inhibition of the Hippo pathway.

The Hippo pathway was originally identified in *Drosophila* by screening for mutations resulting in organomegaly (Xu et al, 1995; Harvey et al, 2003; Huang et al, 2005; Edgar, 2006; Camargo et al, 2007; Dong et al, 2007). The pathway is conserved in mammals and regulates organ size and maintains tissue stem cells (Dong et al, 2007; Pan, 2007; Varelas et al, 2008; Alarcon et al, 2009; Lian et al, 2010; Yimlamai et al, 2014). The core pathway is composed of serine–threonine kinases, mammalian Ste20-like protein kinase 1 and 2 [MST1/serine-threonine kinase 4 (STK4)] and [MST2/serine-threonine kinase 3 (STK3)] and the large tumour suppressor kinase 1 and 2 (LATS1) and (LATS2), the scaffold protein Salvador (Sav1) and co-factor Mob1. Its main downstream effectors are the transcription co-activators Yes-associated protein (YAP) and transcriptional co-activator with PDZ-binding motif (TAZ) (Kanai et al, 2000; Huang et al,

1   Department of Clinical Research, Visceral Surgery and Medicine, University of Bern, Bern, Switzerland
2   Interfaculty Bioinformatics Unit and Swiss Institute of Bioinformatics, University of Bern, Bern, Switzerland
3   Institute of Pathology, University of Bern, Bern, Switzerland
4   Department of Molecular Biology, University of Geneva, Geneva, Switzerland
    *Corresponding author. Tel: +41 31 632 2748; E-mail: deborah.stroka@dkf.unibe.ch

2005). When the canonical Hippo pathway is active, MST1/2 phosphorylates Sav1 facilitating its interaction and the phosphorylation of LATS1/2 (Wu *et al*, 2003). Mob1 interacts directly with LATS promoting LATS auto-phosphorylation. Active p-LATS1/2 phosphorylates YAP1 and TAZ, which facilitates their binding to 14-3-3 restricting them to the cytosolic fraction (Basu *et al*, 2003). Of note, it has been proposed that in the liver YAP1 phosphorylation is not solely dependent on p-LATS1/2 and can involve other intermediate kinases downstream of MST1/2 (Zhou *et al*, 2009). When the Hippo pathway is inactivated, YAP and TAZ are maintained in the nucleus and bind with transcription factors such as p73, PAX3, PPARγ, RUNXs, SMAD, TBX5, PEBP2α, ErbB4 and their primary binding partners, TEAD proteins (Yagi *et al*, 1999; Strano *et al*, 2001; Vassilev *et al*, 2001; Basu *et al*, 2003; Komuro *et al*, 2003; Hong *et al*, 2005; Murakami *et al*, 2006; Zhao *et al*, 2008a,b; Alarcon *et al*, 2009). Genes regulated by YAP activation are those that are involved in cell proliferation and tissue growth, among which are *ctgf, cyr61, Birc5, Ccnb1, Foxm1* and *Areg* (Dong *et al*, 2007; Zhao *et al*, 2008b; Zhang *et al*, 2009; Xin *et al*, 2011; Mizuno *et al*, 2012).

In transgenic models, the hepatic specific deletion of the Hippo kinases, MST1/2, provokes hepatocyte proliferation and liver overgrowth (Zhou *et al*, 2009; Lu *et al*, 2010; Song *et al*, 2010) and a similar outcome is obtained with liver-specific YAP1 overexpression (Camargo *et al*, 2007; Dong *et al*, 2007). Here, we further investigated whether proteins in the Hippo pathway are regulated by regenerative signals following PH and play a physiological role in the control of liver size. We describe that the expression and function of proteins in the Hippo pathway are differentially regulated in young compared to aged livers following partial hepatectomy and support that targeting its core kinases MST1/2 with siRNA is a potential therapeutic intervention to improve the regenerative capacity of resected livers.

## Results

### Regulation of Hippo and YAP1 during liver regeneration

Partial hepatectomy (PH) in rodents is a well-established, robust model to study the complex processes involved in liver regeneration (Higgins & Anderson, 1931). We questioned how the Hippo pathway is regulated during liver regeneration by assessing the expression and activation of the core kinases MST and LATS as well as their effector protein YAP1 in C57Bl/6J mice following PH. Ki67 immunostaining was used to identify hepatocytes re-entering cell cycle. Less than 2% of hepatocytes were Ki67 positive during the hypertrophic phase (0–24 h post-PH) of liver regeneration (Miyaoka *et al*, 2012), whereas up to 75% of hepatocytes became positive during the proliferative phase (24–48 h post-PH) with peak Ki67 expression around 40 h (Fig 1A). Interestingly, we observed an increase of the active, phosphorylated form of MST and LATS (p-MST1/2 and p-LATS1) 6 and 24 h post-PH, which coincides with the non-proliferative, hypertrophic phase of the liver following PH (Fig 1B). As the liver advanced into the proliferative phase (24–48 h post-PH), levels of p-MST1/2 and p-LATS1 decreased to steady state levels. No significant changes in MST1 protein were detected post-PH and no conclusion could be drawn on MST2 due to the high background of the antibody (Appendix Fig S1A and B). Total LATS1 protein (150 kDa) steadily increased 24–48 h post-PH (Fig 1B and Appendix Fig S1A) supporting its role in a YAP-induced feedback mechanism regulation the Hippo pathway (Moroishi *et al*, 2015). The antibody used to detect active MST recognizes both p-MST1 and p-MST2 at Thr183 and Thr180, respectively, and detected the cleaved protein as a 34 and 27 kDa fragment (Praskova *et al*, 2004) (Fig 1B). The band identified as p-LATS1 (Ser909) migrated at approximately 150 kDa which was confirmed in a siRNA knockdown experiment (Appendix Fig S2).

There was no significant transcriptional change in *Mst1, Mst2* or *Lats1* mRNA at 6, 24 or 48 h post-PH (Appendix Fig S3A). Activation of the Hippo effector protein YAP1 was evident 48 h post-PH by its increased expression in nuclear-enriched proteins isolated from liver tissue (Fig 1C) and by its positive nuclear staining in hepatocytes (Fig 1D). To provide further evidence that the parenchymal cells are in part responsible for the increase of YAP protein following PH, we demonstrated that YAP and TAZ are both detectable in isolated cultures of hepatocytes and cholangiocytes (Fig EV1). In addition, there is an increase of YAP expression in hepatocytes isolated from regenerating livers 48 h after surgery compared to sham-operated controls (Fig 1E). YAP activation was further confirmed by an up-regulation of its target genes, *Birc5* (68-fold), *Foxm1B* (80-fold) and the mitotic cyclin, *Ccnb1* (27-fold) 48 h post-PH (Fig 1F). There was no significant change of the YAP target gene *Ctgf* (Appendix Fig S3B)

**Figure 1.  Regulation of the Hippo pathway and YAP during liver regeneration.**

A   Representative photomicrographs of mouse liver sections following a two-thirds PH of Ki67-positive hepatocytes (brown) visualized by IHC. Images are representative of three independent animals per time point. Scale bars, 10 μm.

B   Western blot detection of p-MST1, MST1, p-LATS1, LATS1 from proteins isolated from homogenized liver tissue at several time points. TBP was used as a loading control. Results of a single experiment with *n* = 3 animals per group. Each band is from an independent animal. Protein levels were normalized to TBP and mean value of the ratio of p-MST to total MST and p-LATS to total LATS is plotted.

C   Nuclear-enriched protein fractions obtained from sham-operated livers and liver 48 h post-PH were analysed by Western blot for YAP expression. Sp1 was used as a loading control. Image is representative of a single experiment with *n* = 3 animals per group.

D   Photomicrograph of YAP1 expression in sham and regenerating liver 48 h post-PH visualized by immunofluorescence. Images are representative of three independent animals per condition.

E   Livers from sham and 48 h following PH were digested to isolate the hepatocyte population. Protein extracts were analysed by Western blot for YAP and TAZ expression. β-actin was used as a loading control. Results of a single experiment with *n* = 2 animals per group. Each band is representative of an independent animal.

F   RT–PCR analysis of RNA isolated from mouse liver for *Birc5, Foxm1B* and *Ccnb1*. Data are presented as log$_2$ fold change and were calculated using non-operated mouse liver tissue as a control. Unpaired, two-tailed Student's *t*-test was used to determine the significance between the log$_2$ values of livers 6, 24 and 48 h post-PH compared to the sham-operated controls. Log$_2$ values are plotted on a linear scale as mean ± SD.

Source data are available online for this figure.

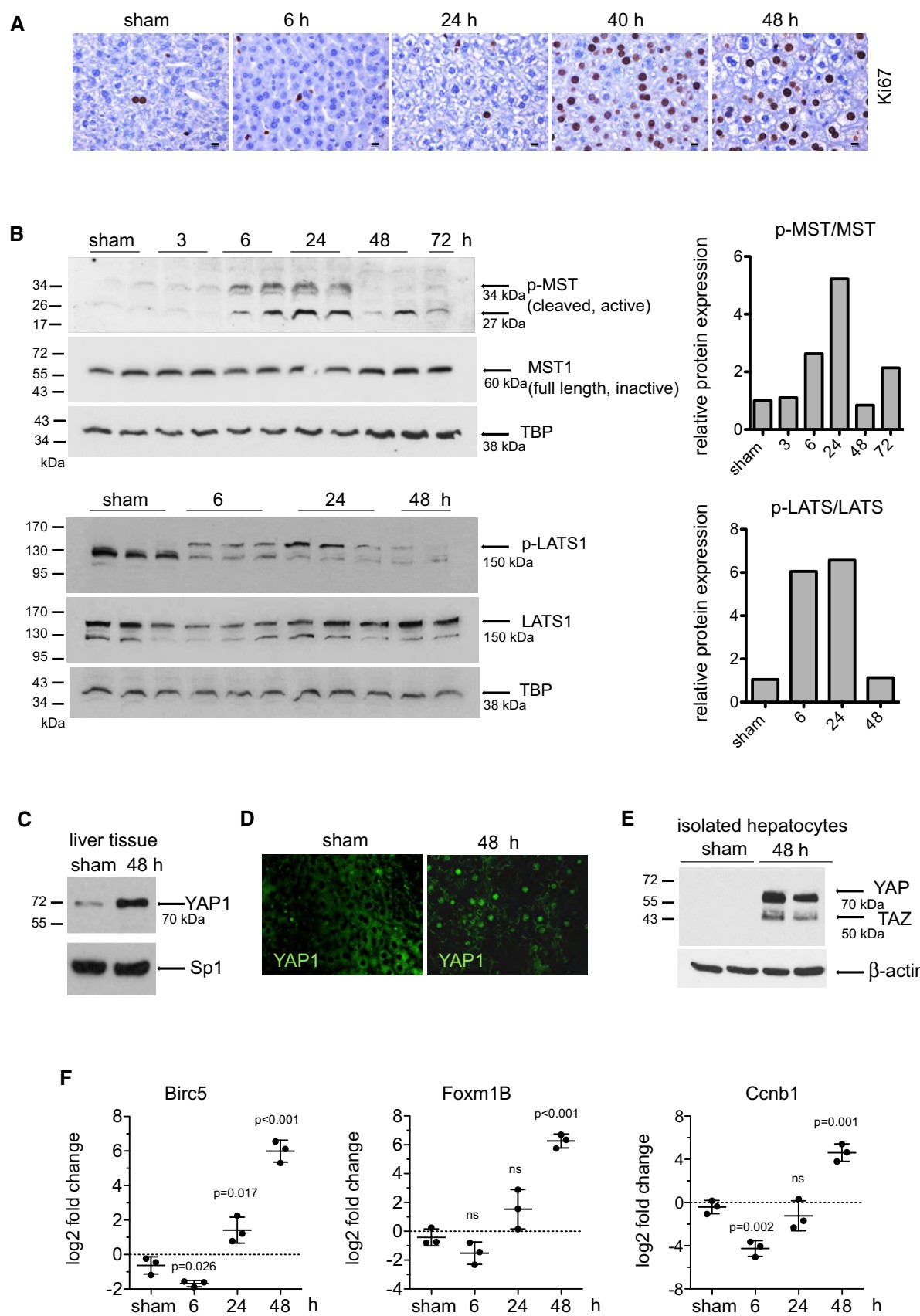

Figure 1.

or of *Yap1* itself, whereas there was significant modulation of *Taz* mRNA expression post-PH (Appendix Fig S3C). These data provide evidence that the Hippo kinases are regulated during the events following PH and YAP is active in regenerating livers.

## Hepatocyte proliferation, Hippo signaling and YAP1 activation are impaired in aged mice

We next assessed for age-related differences in the regulation of Hippo pathway proteins by performing a 60% PH in aged (> 12 months old) and young (6–8 week) mice. A fixed end time point of 40 h was set in order to analyse liver tissue at the peak of proliferation (Wang *et al*, 2001). Compared to 100% (6/6) of the young mice, only 66% (6/9) of aged mice survived up to 40 h within the parameters defined by our ethical score sheet. The non-surviving aged mice recovered from surgery, but were severely strained 24 h post-PH and required euthanization. The livers of euthanized mice were determined to be not regenerating and were excluded from further analysis. The proliferation index (p.i.) of the remaining six aged and six young animals was calculated by the ratio of Ki67-positive hepatocytes to total number of hepatocytes and a p.i. of < 10% was used as a cut-off to classify the livers as (−) non-regenerating, while a p.i. of 10 or greater signified (+) regeneration (Fig 2). The livers of all young mice 6/6 were regenerating (p.i. = 75 ± 10; Fig 2A) vs. only 2/9 of the aged livers (p.i. = 37.5 and 30.7; Fig 2B). The p.i. of the remaining four aged mice scored < 10 (p.i. = 4.2 ± 2.4; Fig 2C) and therefore was classified as non-regenerating (Fig 2B). Histologically, all regenerating livers showed discreet architectural changes, and young livers had slight microvascular steatosis (Fig 2A), whereas aged regenerating livers had marked steatosis (Fig 2B). Aged non-regenerating livers had signs of foamy degeneration of the hepatocytes and virtually no hepatic steatosis (Fig 2C).

Compared to young livers, aged livers had higher steady state levels of pMST that remained elevated during regeneration (Fig 3A). Steady state level of pLATS was also higher in aged animals and was not modulated following PH (Appendix Fig S4). Total MST1 protein decreased 40 h post-PH in young livers, but remained elevated in aged mice. No difference was observed in levels of p-YAP between young and aged animals. However, total YAP protein increased in young animals following PH, whereas in aged animals there was a higher steady state level that was not altered following PH (Fig 3A). This observation was unique to YAP as TAZ expression increased in a similar manner in both young and aged animals (Fig 3A). Moreover, there was no increase of YAP1 targets *Foxm1B*, *Birc5*, *Ccnb1* or other regulators of cell cycle such as, *Ccna2* and *Ccnd1* and a tendency of higher Cdkn1A levels in non-regenerating aged animals (Fig 3B). Taken together, these data support that there are age-related defects in liver regeneration following PH and regulation of the Hippo pathway is anomalous in non-regenerating aged livers.

## Inhibition of MST1/2 with RNAi provokes hepatocyte proliferation *in vivo*

Bringing together our observations presented above with data that has shown MST1/2 are constitutively activated in a quiescent liver (Zhou *et al*, 2009; Lu *et al*, 2010; Song *et al*, 2010), we chose to inhibit MST1/2 by RNAi as a potential means to improve the liver's regenerative potential. Both MST1 and MST2 were targeted due to their reported redundancy and overlapping function (Zhou *et al*, 2009; Lu *et al*, 2010). We screened and selected siRNA sequences based on their ability to inhibit MST1 and MST2 expression in a mouse liver cell line (Appendix Fig S5). For *in vivo* delivery, siRNAs were encapsulated with liposomes and delivered by femoral vein (i.v.) injection. In control experiments, we confirmed that the liver and more specifically hepatocytes were targeted with lipid-encapsulated siRNA by knocking down factor VII (FVII), a coagulation protein synthesized specifically by hepatocytes. Venous injection of lipid-encapsulated siRNA targeting FVII achieved an 80% reduction of *FVII* mRNA in the liver and 90% reduction of circulating FVII protein (Fig EV2).

In young mice, knockdown of MST1/2 with siRNA (siMST) effectively reduced hepatic expression of *Mst1* & *Mst2* to 15 and 45%, respectively, within 24 h after i.v. injection. The efficiency of the knockdown declined over time but still partially remained 6 days after injection (Fig 4A). Concordant with the loss of *Mst1* mRNA, MST1 protein was depleted and lower levels of p-LATS1 were detected over time (Fig 4B). In siMST livers, there was an increase of YAP1 and a decrease of p-YAP1 in nuclear-enriched protein extracts 3 days after injection (Fig 4C). The amount of nuclear proteins was normalized with histone H3, which remained constant; however, its phosphorylated form (p-H3), a marker of mitosis, was increased in siMST-injected livers (Fig 4C). There was a significant up-regulation of YAP1 target genes, *Birc5* (2.8 ± 0.2-fold), *Foxm1B* (5.3 ± 0.3-fold) and *Ccnb1* (6.8 ± 0.6-fold) at day 3 and *Foxm1B* (5.3 ± 0.3-fold) and *Ccnb1* (6.8 ± 0.6-fold) at day 6 (Fig 4D). The proliferation marker, Ki67, was positive in the hepatocytes of the KD tissue, whereas the biliary epithelial cells/bile ducts were negative, favouring a hepatocytic over ductal response (Fig 4E). The percentage of proliferating cells was calculated by the number of Ki67-positive cells relative to the total number of hepatocytes measured by nuclear size and HNF4-α staining. siMST provoked a hepatocyte p.i. of 5.7 ± 1.2 at day 3 and 6.9 ± 1.5 at day 6 post-injection, whereas the p.i. of the scrambled controls (siScr) was 1.1 ± 0.8 (Fig 4E). We observed no increase of liver-to-body weight ratio up to day 6 post-injection.

For a broader view on the effect of siRNA targeting MST1/2, we used a gene expression microarray to compare the transcriptome of livers treated with siMST with siScr and untreated control livers. Few individual genes were significantly regulated between the siMST and siScr groups; however, a gene set enrichment analysis (GSEA) performed with the algorithm SetRank revealed several affected pathways (Appendix Fig S6). The first group of genes had known antiviral activity such as the 2′–5′ oligoadenylate synthetases Oas2, Oas1a, Oasl1 and Oas1g and several MHC II genes such H2-D1, H2-K1, H2-Q4 and H2-Q7 are related to interferon gamma signaling. They were up-regulated when comparing the siMST livers to both the siScr and untreated livers. Although this specific antiviral response is not observed when comparing the siScr group to untreated livers, it is still likely to constitute a reaction of the liver cells to treatment with dsRNA, as we observe a different antiviral response in the siScr livers (Appendix Fig S7A). The second group related to cell cycle, more specifically with mitotic prophase. Livers treated with siMST showed an up-regulation of several elements of the kinetochore complex such as Kif2c, Nuf2 and Ndc80, along with key cell cycle activators such as Cdk1 and Ccnb2 (Appendix Fig S7B). The third group consists of gene sets related to the

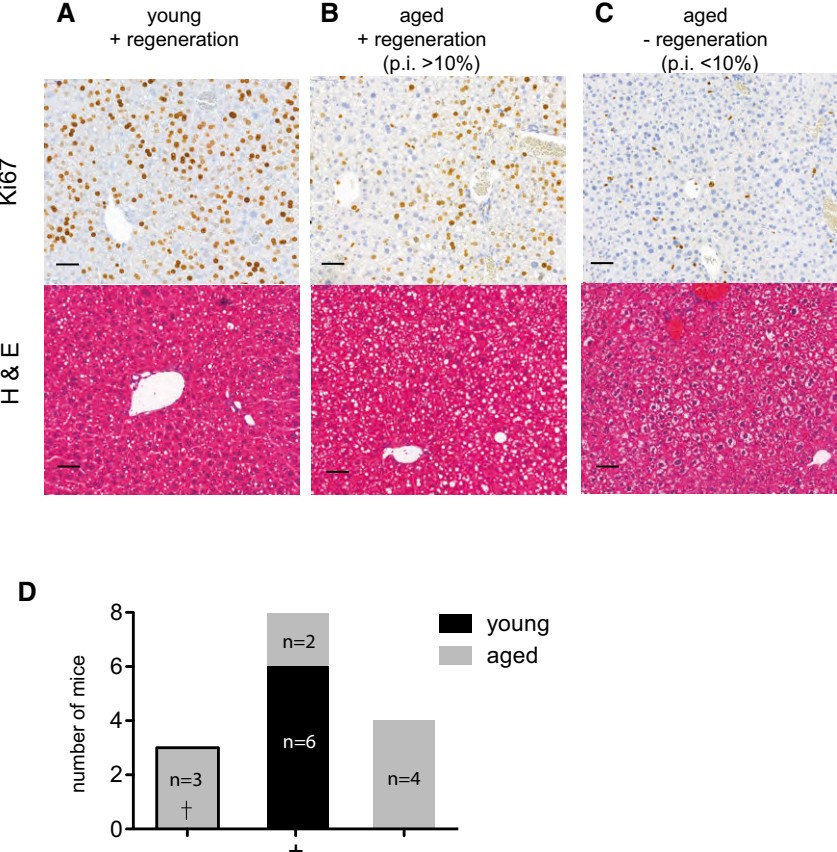

**Figure 2. Hepatocyte proliferation is impaired in aged mice.**

A–C  Representative photomicrograph of young (6–8 weeks) and aged (> 12 months) livers 40 h post-PH stained with Ki67 and visualized by IHC and H&E for histology. Sections were scanned using 3DHistech Pannoramic MIDI Scanner and Ki67-positive cells were quantitated with Quant Center 2.0 software. A proliferation index (p.i.) was calculated based on the number of Ki67-positive hepatocytes to total hepatocytes and livers were classified as (+) regenerating, p.i. > 10% or (−) non-regenerating, p.i. < 10%. Scale bars, 50 μm.

D  Number of young (6–8 weeks) and aged (> 12 months) mice with (−) non-regenerating or (+) regenerating livers based on p.i., 40 h post-PH. Animals deemed non-surviving for ethical reasons were euthanized 24 h post-PH. Representative results from a single experiment with *n* = 9 aged and *n* = 6 young animals per group.

extracellular matrix (EM) such as the collagen genes Col1a1, Col1a2 and Col3a1, the laminin genes Lamb1, Lamb2 and Lamc1, fibulin 5 (Fbln5), fibrin 1 (Fbn1), heparan sulphate proteoglycan 2 (Hpgn2) and lumican (Lum) (Appendix Fig S7C). The detection of gene sets involved in cell proliferation and formation of matrix suggests that transient inhibition of MST1/2 is permissive for organ growth. Finally, analysis of the serum from control or siMST-injected animals revealed no increase of alanine transaminase (ALT) or aspartate transaminase (AST), suggesting that liver damage did not account for the increased hepatic proliferation (Appendix Fig S8A). In addition, there was no significant decrease of *Mst1* or *Mst2* mRNA expression in the lung and spleen indicating low, remote off-target effects (Appendix Fig S8B).

## Inhibition of MST1/2 improves liver regeneration in aged mice

We next tested the hypothesis that inhibition of MST1/2 could rescue an age-related defect in liver regeneration. Aged mice were injected with either siRNA targeting MST1/2 (siMST) or a scrambled control (siScr) and 24 h after injection a 70% PH was performed. The resected liver tissue was used to determine the efficiency of the KD and siMST1/2 effectively knocked down hepatic expression of *Mst1* & *Mst2* to 18 and 55%, respectively (Fig EV3). Forty hours post-PH, there was a 0% (0/9) survival rate of the aged mice injected with siScr, whereas there was 66% (6/9) survival of aged mice injected with siMST (Fig 5A). The livers of siMST animals scored with a mean p.i. of $36.17 \pm 11.18$ (Fig 5B). Ki67-positive cells were observed in the hepatocytic compartment of the livers only; there were no signs of ductal reactions or oval cell expansion as determined by H&E stain, CK-19 expression and Ki67 (Fig 5C). As an additional sign of regeneration, *Birc5, Foxm1B* and *Ccnb1* were significantly up-regulated 40 h post-PH in the aged siMST-treated livers (Fig 5D). When comparing the regeneration of livers treated with siMST to the non-treated aged controls, knockdown of MST1/2 significantly increased the p.i. (aged: $14.2 \pm 15.7$ vs. siMST: $36.2 \pm 11.2$; Fig 5E) and increased the liver-to-body weight ratio (Fig 5F) demonstrating an overall advantage of targeting MST in aged animals. To verify that cellular hypertrophy was not the

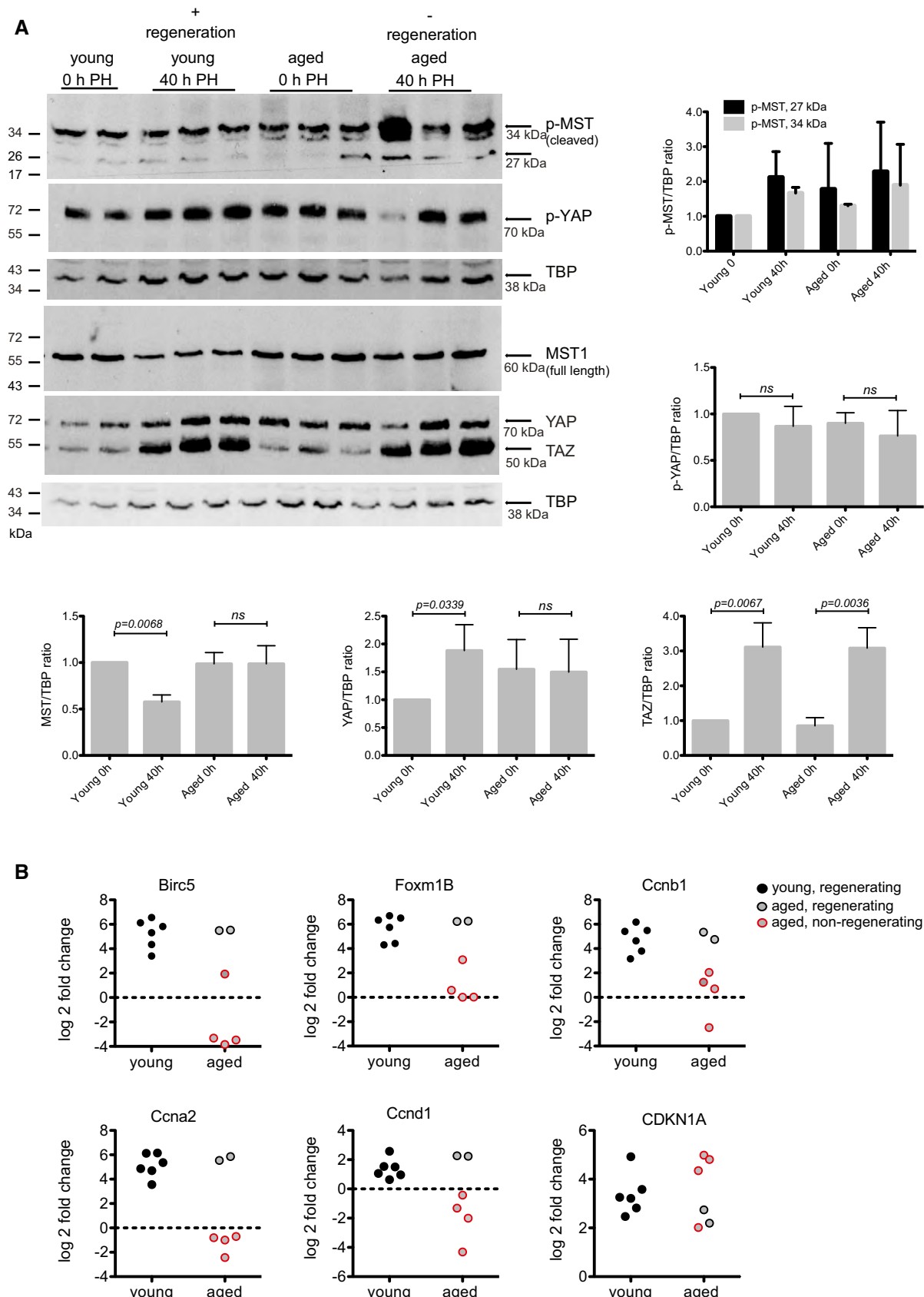

**Figure 3.**

**Figure 3.  Hippo signaling and YAP activation are impaired in aged mice.**

A   Western blot detection of p-MST1, MST, p-YAP, YAP and TAZ in control and 40 h following PH in young and aged mice. TBP was used as a loading control. Representative results from a single experiment with *n* = 6 independent animals. Each band is from an independent animal. Western blot quantification ratios are graphically depicted. Protein levels were normalized to TBP and mean value of the group is plotted. Unpaired, two-tailed Student's *t*-test was used to calculate the significance between 0 h (resected liver) and 40 h following PH. Bars represent mean ± SD.

B   RT–PCR analysis of RNA isolated from mouse liver for *Foxm1B, Birc5* and *Ccnb1, Ccna2, Ccnd1* and *Cdkn1A*. RT–PCR data are presented as log$_2$ fold change and each value post-PH was calculated for each mouse by comparison with its resected liver lobe. Circle with red outline marks the aged non-regenerating livers as shown in Fig 2B. Each dot is representing an independent animal.

Source data are available online for this figure.

cause of the increased liver-to-body weight ratio, the geometric diameter of the hepatocytes was measured (Appendix Fig S9). There was an increase of hepatocyte size in regenerating livers of both young and aged animals. However, no significant change in the size of regenerating hepatocytes was found in aged compared to aged + siMST-treated animals (Fig 5G). This suggests that cellular hypertrophy is not the cause of the increased liver-to-body weight ratio in the aged + siMST-treated animals, but rather an increase of hepatocyte size and number. And finally, serum ALT, AST and alkaline phosphatase (ALP) levels were elevated following PH, indicating liver cell injury in aged animals; however, there were no significant differences in regenerating aged mice treated or not treated with siMST (Appendix Fig S10). Taken together, using an aged mouse model, we demonstrate the clinical applicability of targeting MST1/2 to improve the liver's regenerative capacity.

# Discussion

Our data demonstrate that the liver's physiological growth response following PH involves MST1 and LATS protein modification and YAP1 activation, which is demonstrated by its nuclear localization and transcriptional activation of target genes. YAP1 involvement in liver regeneration is in agreement with others (Wu *et al*, 2013; Grijalva *et al*, 2014); however, our data fine-tune the role of Hippo during the early hypertrophic and proliferative phases of the regenerative response. Phosphorylated MST1 and LATS1 were detected in quiescent livers; however, their phosphorylation increased at early time points after PH. Although this may appear contradictory to a proliferative response, this increase occurs during the hypertrophy

phase of regeneration before the onset of hepatocyte proliferation (Miyaoka *et al*, 2012). As the hypertrophy phase transitions into the proliferative phase, levels of p-MST and p-LATS1 decrease coinciding with Ki67 immunostaining and YAP1 activation. We did not observe phosphorylated MST1 or LATS1 following PH to be reduced below their steady state levels, allowing us to speculate as to the importance of the flux of their expression in the control of YAP1. Moreover, one can speculate that the steady state levels of pMST and pLATS needed to maintain YAP in a quiescent liver are not sufficient to control its expression when the regenerative signals are fully engaged, as occurring following PH. The increase of p-LATS1 coincided with the increase of its upstream regulator p-MST1, which is supported by reports showing significant decrease of LATS1 phosphorylation following inactivation of MST1/2 (Lu *et al*, 2010). However, it was also demonstrated that MST proteins could regulate YAP1 independent of LATS1 (Zhou *et al*, 2009). Therefore, although we observed a strong regulation of MST1 and LATS1 protein during liver regeneration, we cannot conclude by which mechanism YAP1 is being controlled.

Several liver-specific knockout models targeting Hippo pathway proteins such as MST1/2 (Zhou *et al*, 2009; Lu *et al*, 2010; Song *et al*, 2010), NF2 (Benhamouche *et al*, 2010; Zhang *et al*, 2010) and WW45 (Lee *et al*, 2010) have been generated. Notably, there is the target-dependent effect in the liver. A liver-specific KO of MST1/2 resulted in hepatocyte proliferation, whereas NF2 depletion mainly targeted biliary cells, and WW45 KO increased the number of hepatic progenitor cells. As hepatocyte proliferation is fundamental for liver regeneration, MST1/2 presents itself as an optimal therapeutic target. Moreover, compared to structural proteins, kinases are easier targets for potential pharmacological intervention. Until

**Figure 4.  Inhibition of MST provokes hepatocyte proliferation *in vivo*.**

A   Eight-week-old mice were injected with liposomes coupled to scrambled siRNA (siScr) or sequences targeting MST1 and MST2 (siMST). Each dot is representing an independent animal. The amount (%) of *Mst1* and *Mst2* mRNA remaining in the liver was tested by RT–PCR 1, 3 and 6 days post-injection and calculated compared to control non-transfected livers. Unpaired, two-tailed Student's *t*-test was used to calculate the significance of mRNA remaining in comparison with non-transfected control livers at each time point.

B   Western blot detection of MST1, p-LATS1 and β-actin 1, 3 and 6 days post-siRNA injection. A representative animal from each time point is shown.

C   Western blot detection of YAP1, p-YAP1, H3 and p-H3 3 days post-siRNA injection. Two representative animals from each group are shown. All samples were run on the same blot. The blot is split due to non-consecutive loading of the samples.

D   Quantitative RT–PCR analysis of RNA isolated from mouse liver for *Foxm1B, Birc5* and *Ccnb1*, 1, 3 and 6 days post-injection. Log$_2$ fold change was calculated using non-transfected mouse liver as control. Unpaired two-tailed Student's *t*-test was used to calculate the significance of fold change of in comparison with a panel of non-transfected control livers.

E   HNF4α and Ki67 were detected by IHC and cells positive for both were classified as proliferating hepatocytes. Ki67-positive hepatocytes are indicated with an asterisk and Ki67-negative bile ducts/biliary cells with an arrow in tissue 3 days post-siRNA delivery. The percentages of positive hepatocytes at day 3 and day 6 post-injection are graphed. Two-tailed Student's *t*-test was used to calculate the significance of percentage of positive hepatocytes in comparison with non-transfected control livers.

Data information: Throughout the figure, representative results from two independent experiments with *n* = 3 animals per group are depicted. Bars represent mean ± SD.

Source data are available online for this figure.

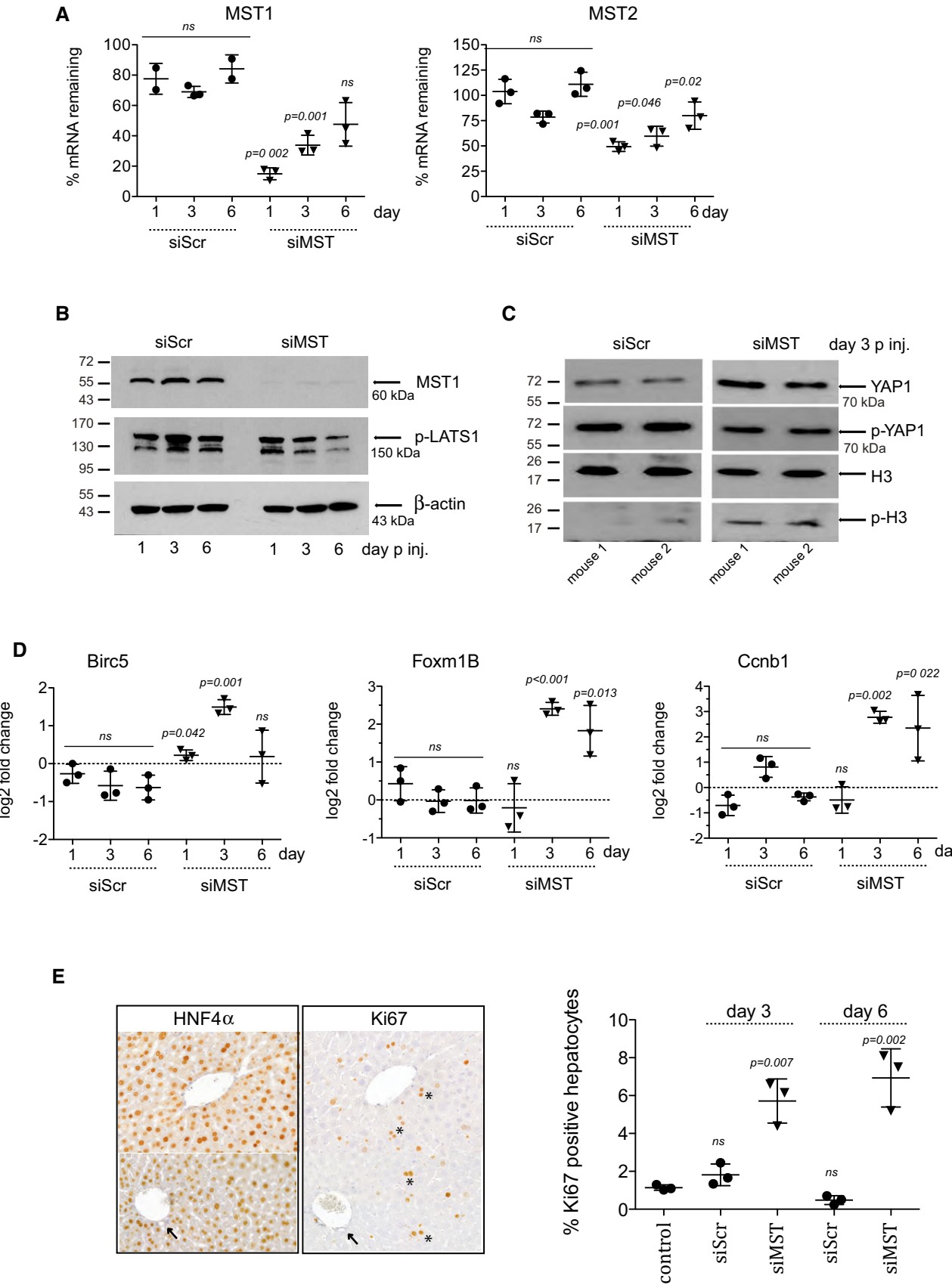

**Figure 4.**

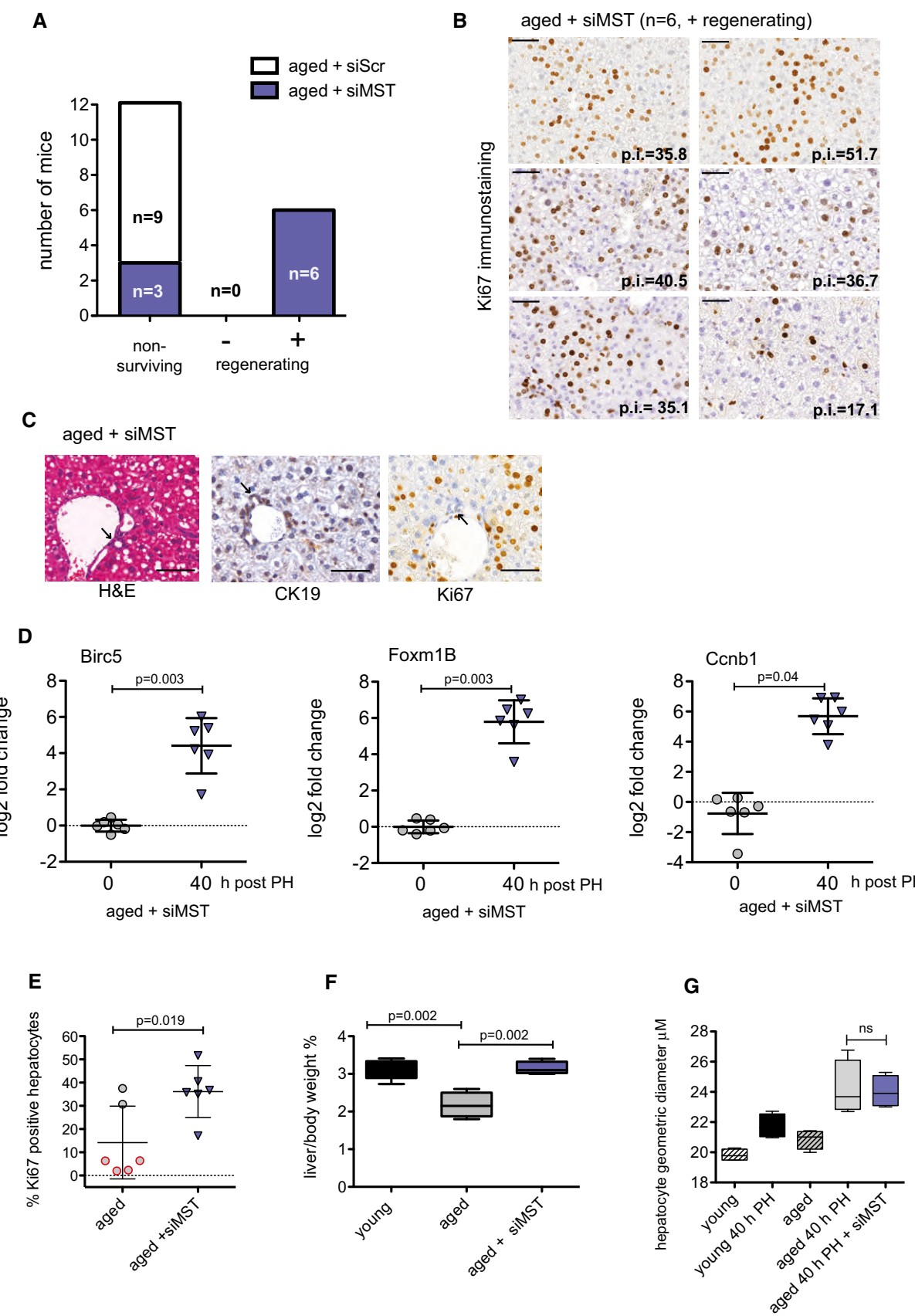

Figure 5.

**Figure 5.  Inhibition of MST improves liver regeneration in aged mice.**

A   Aged mice (> 12 months) were injected i.v. with siScr or siMST. Twenty-four hours after siRNA injection, a 70% PH was performed and remnant/regenerating liver tissue was harvested 40 h later. p.i. of the livers was calculated by IHC staining of Ki67-positive hepatocytes and livers were classified as (−) non-regenerating or (+) regenerating based on the percentage of positive cells. Non-surviving animals were sacrificed before the 40-h end time point. "*n*" indicates the number of animals per group.

B   Ki67 immunostaining of the regenerating aged mice livers, 40 h post-PH. Sections were scanned using 3DHistech Pannoramic MIDI Scanner and quantitated with Quant Center 2.0 software. p.i. values are provided on the image.

C   Representative photomicrograph of an aged liver treated with siMST and stained with H&E and by IHC for CK19 and Ki67. Arrows indicate ductal regions with no signs of reaction or oval cell expansion.

D   Quantitative RT–PCR analysis of RNA isolated from mouse liver for *Foxm1B*, *Birc5* and *Ccnb1* at 0 and 40 h post-PH. Log$_2$ fold change was calculated using non-transfected/non-resected mouse liver as a control. Representative results from a single experiment with *n* = 6 animals per group are shown. Paired, two-tailed Student's *t*-test was used to calculate the significant change in signals between liver tissues at the time of resection (0 h) compared to 40 h post-PH.

E   Percentage of Ki67-positive hepatocytes in aged and aged + siMST 40 h post-resection. Representative results from a single experiment with *n* = 6 independent aged animals for group are shown. Two-tailed Student's *t*-test was used to calculate the significance of percentage of positive hepatocytes of siMST-treated animals compared to non-treated controls.

F   Endpoint liver weight was taken of the remnant lobe and expressed as a percentage to total body weight. Representative results from a single experiment with *n* = 6 independent animals for group expect for control group where *n* = 6 are shown. Estimated liver-to-body weight ratio before PH in mice = 3.85 ± 0.05 (SD). Unpaired, two-tailed Student's *t*-test was used to calculate the significance of each aged group in comparison with the control young group.

G   Hepatocyte geometric diameter was determined (see Materials and Methods). Representative results from a single experiment with *n* = 3 independent animals per group. Unpaired, two-tailed Student's *t*-test was used to calculate the significant change in hepatocyte size in liver tissue with and without siMST 40 h post-PH.

Data information: Bars represent mean ± SD. Scale bars, 50 μm.

now, no data are published on the effect of a liver-specific LATS1 and 2 deletion. Recently, however, a liver-specific Lats2 single-knockout (Lats2flox/flox; Albumin-Cre, Lats2-cKO) was reported to show no overt histological abnormalities nor developed liver tumours up to 16 months of age (Park *et al*, 2016). MST1/2 regulation of YAP1 is reported to be tissue specific, particularly in the liver (Zhou *et al*, 2009). Thereby, kinases other than LATS may be responsible for YAP1 phosphorylation downstream of MST1/2, further supporting MST proteins as suitable targets.

We chose to use siRNAs as the means to genetically target MST1/2. The temporary effect offered by siRNA is attractive in order to minimize the chance that inhibition of Hippo signaling could provoke liver overgrowth or hepatocyte transformation. Using siRNA coupled with liposomes has the advantage of reaching our target cell the hepatocyte with higher specificity, thereby reducing the effect of YAP1 activation in other cell types. The liposomes used in this study successfully reached the hepatocyte and delivered the siRNA in a sufficient concentration to knock down our targeted genes of interest. In young mice, this appeared to be achieved in the absence of any additional stress/injury to the liver, as we did not observe an increase of serum transaminase. Also, mice injected with control siRNA to a scrambled sequence showed no significant signs of proliferation. However, the poor recovery following PH of aged animals receiving encapsulated control siRNA suggests that the liver is stressed; however, this effect is most likely due to the liposome delivery, as the siRNA targeting MST1/2 improved regeneration, even compensating for the delivery system. Specific targeting of MST1 or 2 should not result in liver injury as serum transaminases are not elevated in MST1/2 double mutants (Lu *et al*, 2010).

Targeting MST1/2 with encapsulated liposomes reduced MST1/2 mRNA and MST1 protein, induced YAP1 activation and provoked hepatocytes to re-enter cell cycle. YAP1 targets genes involved in cell proliferation were increased, whereas CTGF was not increased in mouse (Appendix Fig S3) or rat (Grijalva *et al*, 2014) liver following PH. Targeting MST1/2 with siRNA did result in a mild phenotype, which may be viewed as a desired effect to avoid detrimental side effects of inhibiting Hippo in the liver, or overexpressing YAP1

directly, such as hepatocyte overproliferation or de-differentiation (Yimlamai *et al*, 2014). While our manuscript was under review, a study reported siRNA nanoparticles targeting MST1, MST and Nf2 results in extensive proliferation of hepatocytes and activation of YAP1 (Yin *et al*, 2016). In their model, hepatomegaly was only observed in the triple knockdown, 15 days following repeated injections. This in part explains the mild phenotype we observed in quiescent liver targeting only MST1 and MST2 after 6 days with a single siRNA injection and advocates optimization of treatment protocols. Nevertheless, both studies support the Hippo pathway as a viable therapeutic target to manipulate liver growth.

By restoring regeneration in an aged mouse model, we were able to demonstrate the potential clinical applicability of targeting the Hippo pathway. Several groups have reported aged-related defects in liver regeneration (Bucher *et al*, 1964; Fry *et al*, 1984; Wang *et al*, 2002; Pibiri *et al*, 2015) and YAP1 activation was suggested to help aged mice recover after PH (Pibiri *et al*, 2015). The beneficial effect we observed with siMST could be a result of a direct activation of YAP1 in the targeted hepatocytes. Foxm1B expression is restricted to proliferating cells and strongly up-regulated upon entry into the S phase of cell cycle (Korver *et al*, 1997) and was increased in liver injected with siMST. In a report by Wang *et al*, mice with sustained hepatic Foxm1B overexpression were able to overcome age-related defects during liver regeneration; however, Foxm1B alone is insufficient to induce differentiated cells to enter cell cycle (Ye *et al*, 1999; Wang *et al*, 2002). *Birc5* and *Ccnb1* were also increased by MST1/2 RNAi. However, as MST1 is involved in other signaling pathways including apoptotic signaling (Ardestani *et al*, 2014; Del Re *et al*, 2014; Chao *et al*, 2015), it is not conclusive whether the effect we achieve is due only to YAP1 activation. This question is prompted by our observations in the aged mice, in which siMST-treated animals showed a survival advantage at 24 h compared to the siScr control, a time point prior to the actual proliferative phase of the regenerative response.

The Hippo pathway is becoming an attractive target in the liver for therapeutic intervention in the field of hepatic malignancies (Fitamant *et al*, 2015; Yimlamai *et al*, 2015) and hepatic re-growth

(Yin *et al*, 2016). As Hippo is also important for tissue stem cell maintenance (Lu *et al*, 2010), these implications may reach beyond the liver and to the repair and regeneration of other organs. Research and development of small molecule kinase inhibitors that modulate the activation of Hippo kinases or antisense drugs to silence expression of key regulatory proteins hold promising therapeutic and investigative potential in regenerative medicine.

# Materials and Methods

### Partial hepatectomy model

The mice used in this study were 6- to 8-week- (~20 g) and 12-month-old (~29 g) female C57BL/6JRccHsd provided by Harlan, the Netherlands. Animals were kept in a temperature-controlled room with a 12-h dark/light cycle. Experiments were done with Institutional Animal Care and Use Committee approval and in strict accord with good animal practice as defined by the Office of Laboratory Animal Welfare. Littermates were randomly assigned to control sham or partial hepatectomy groups. Mice were anesthetized by isoflurane inhalation during an operation time of 15–20 min. For analgesia, Temgesic (0.05 mg/kg) was injected i.p. prior to surgery. Two-thirds partial hepatectomy (2/3 or 70%) was performed according to methods previously described (Inderbitzin *et al*, 2006). Briefly, following a laparotomy, vicryl 4.0 suture (VCP496, Ethicon) was used to ligate and excise the median and left liver lobes. The peritoneal cavity was washed with saline solution and the abdomen was sutured closed. In sham-operated mice, a laparotomy was performed, the liver was manipulated with a cotton-coated stick, flushed with saline, and the abdomen was sutured closed. The resected tissue was collected and used as a non-regenerating matched control for each mouse. Liver tissue from non-operated and sham-operated mice was also used as controls. Liver regeneration was followed for various time points up to 6 days. At experimental endpoints, mice were anesthetized by isoflurane inhalation. Blood was collected via the vena cava and centrifuged for serum. Mice were sacrificed by exsanguination. All excised tissues were cut in 3 × 3 mm pieces, frozen in liquid nitrogen and stored at −80°C or fixed in 4% formalin overnight at room temperature.

### Primary liver cell isolations

Hepatocytes from control or regenerating livers were isolated using a two-step enzymatic perfusion protocol (Portmann *et al*, 2013). The viability of the isolated hepatocytes was determined by trypan blue exclusion, and only preparations of over 90% viability were used. The hepatocytes were seeded onto rat tail collagen-coated tissue culture plastics in Dulbecco's modified Eagle medium containing 10% foetal bovine serum, left to attach for 1–2 h and then washed twice with phosphate-buffered saline (PBS) to remove unattached cells. The hepatocytes were cultured in arginine-free Williams E medium supplemented with insulin (0.015 IU/ml), hydrocortisone (5 μmol/l), penicillin (100 IU/ml), streptomycin (100 μg/ml), glutamine (2 mmol/l) and ornithine (0.4 mmol/l) for 24 h before use.

Cholangiocytes were isolated by enzymatic tissue digestion, differential density centrifugation followed by immunomagnetic selection for CD326 (EpCAM). The cholangiocytes were cultured in DMEM/F12 containing 5% foetal bovine serum, 10 ng/ml hepatocyte growth factor (HGF), 10 ng/ml epidermal growth factor (EGF), 4 μg/ml hydrocortisone, 10 ng/ml cholera toxin, 1.24 i.u./ml insulin, $2 \times 10^{-9}$M triiodothyronine and penicillin–streptomycin.

### siRNA screen in BNL cells

BNL 1ME A.7R.1 (ATCC, TIB-75™) cells were cultured in DMEM with 10% FBS, 100 U/ml penicillin and 100 μg/ml streptomycin (Life Technology) in a humidified incubator at 37°C with 5% $CO_2$. All experiments were carried out at 60–70% confluence. Cells were transfected with siRNA targeting MST1 (also known as STK3) ID: s81598, s81597, s81599 and MST2 (also known as STK4) ID: s211751, s211750, s211749 (Ambion, Life Technologies) using Lipofectamine 2000 (Invitrogen) in DMEM with 10% FBS. After 48 h, total proteins were extracted and analysed by Western blot.

### siRNA LATS knockdown in HEPG2 cell line

HEPG2 cells were cultured in DMEM with 10% FBS, 100 U/ml penicillin and 100 μg/ml streptomycin (Life Technology) in a humidified incubator at 37°C with 5% $CO_2$. All experiments were carried out at 60–70% confluence. Cells were transfected with siRNA targeting LATS1 siGENOME Human LATS1 (9113) siRNA SMARTpool targeting four different sequences D-004632-01 (GAACCAAACUCUCAA AACAA), D-004632-02 (GCAAGUCACUCUGCUAAUU), D-004632-03 (GAAAUCAAGUCGCUCAUGU) and D-004632-02 (GAUAAAGACAC UAGGAAUA) (Dharmacon) using Lipofectamine 2000 (Invitrogen) in DMEM with 10% FBS. After 72 h, total proteins were extracted and analysed by Western blot.

### Protein isolation and Western blot

Total protein extracts were isolated from liver tissue using 500 μl RIPA lysis buffer and disassociated using a TissueLyser (Qiagen) for 2 min at 20 Hz. The liver lysates were centrifuged at 23,000 *g* for 15 min at 4°C, and supernatants were collected. To isolated enriched nuclear protein extracts, liver tissue was crushed with mortar and pestle while being kept frozen in liquid nitrogen and further broken up with a manual homogenizer in cell lysis buffer (10 mM Tris at pH 8.0, 150 mM NaCl, 1 mM EDTA and phosphatase inhibitors phenylmethylsulfonyl fluoride (PMSF), sodium orthovanadate, sodium fluoride). Homogenates were centrifuged for 5 min at 3,200 *g* and the cytoplasmic fraction collected. Pellets were re-suspended in nuclei extraction buffer (20 mM HEPES pH 7.9, 400 mM NaCl, 1 mM EDTA) and after 15 min on ice were centrifuged for 15 min at 15,000 *g*. All protein extraction buffers contained protease inhibitor cocktail (P8340, Sigma). Protein concentrations were determined with the Bio-Rad Protein Assay System (Bio-Rad) as described by the manufacturer.

Equal amount of proteins were separated by SDS–PAGE and transferred onto PVDF membrane (GE Healthcare, Amersham) by wet transfer (Bio-Rad). Membranes were blocked in 5% not-fat dry milk in TBST and incubated with primary antibodies overnight at 4°C and 90 min in the secondary antibody. All primary antibodies were obtained from Cell Signaling unless otherwise noted: MST1 (1:500, #3682), MST2 (1:500, #3952), p-MST1/p-MST2 ((Thr183/

Thr180) 1:500, #3681), LATS1 (1:500, #ab70561, Abcam), p-LATS1 ((Ser909) 1:500, #9157), YAP1 (1:500, #4912), YAP/TAZ (1:1,000, #8418), p-YAP1 ((Ser127) 1:1,000, #4911), H3 (1:2,000, #4499), p-H$_3$ ((Ser10) 1:2,000, #06-570, Upstate Biotechnology), HNF4α (1:500, #SC-8987, Santa Cruz), CK-19 (1:2,000, #NB100-687, Novus Biologicals) monoclonal anti-β-Actin–peroxidase clone AC-15 (1:50,000 #A3854, Sigma), TBP (1:1,000, #8515). Secondary antibody used was anti-rabbit-HRP (1:2,000–1:5,000) (Dako). Signals were visualized by enhanced chemiluminescence (Western Lightning Plus ECL, Perkin Elmer) and films were developed with CURIX 60 (AGFA). The band size was estimated using Page Ruler™ Prestained Protein Ladder (Fermentas) and Precision Plus Protein™ DUAL Color Standards (BIO-RAD, #161-0374). Band densities were analysed using ImageJ software and signals were normalized to TBP or β-actin. The Student's *t*-test was used for comparison of signal intensities.

## Immunohistochemistry

The tissue was processed for paraffin embedding. Paraffin sections were cut at a thickness of 6 μm. Tissues were stained for histological analysis with haematoxylin and eosin. Primary antibodies used were HNF4α (SC-6556, Santa Cruz), YAP1 (#4912, Cell Signaling) and Ki67 (clone SP6, RM-9106-R7 Thermo Scientific) followed by secondary antibodies from Reactolab using Vectastain Elite ABC Kit for goat IgG (PK-6105) or rabbit IgG (PK-6101). For quantification of Ki67-positive hepatocytes in the knockdown experiment without PH, 20 high power fields were photographed per sample and hepatocytes were manually counted for HNF4α- and Ki67-positive reactions. For the quantification of the number of hepatocytes in proliferation in aged and young mice, the 3DHISTECH Pannoramic MIDI Scanner and 3DHISTECH Quant 2.0 software were used. A proliferation index (p.i.) was calculated as a ratio of Ki67-positive nuclei to total number of hepatocytes counted. The results are presented as the p.i. ± SD and were compared with an unpaired Student's *t*-test.

## Quantitative RT–PCR

Total RNA was isolated from all mouse samples and cells by using a TRIzol reagent-based method following the manufacturer's protocol (Life Technologies). 500 ng of total RNA was used for cDNA synthesis using Omniscript RT Kit 200 (Qiagen). mRNA was analysed by quantitative RT–PCR with TaqMan gene expression assays and reagents according to the standard protocols (Applied Biosystem). Mouse probes used include amphiregulin (Mm01354339_m1), Birc5 (Mm00599749_m1), Cdkn1A (Mm004324480), CTGF (Mm011929 33_g1), Ccna2 (Mm00438063_m1), Ccnb1 (Mm01322 149_mH), Ccnd1 (Mm00432359), Foxm1B (Mm00514925_m1), LATS1 (Mm01191886_m1), MST1 (Mm00451755_m1), MST2 (Mm 00490480_m1), TAZ (Mm00504978_m1), YAP1 (Mm01143263_m1), 18S (Mm02601777_g1) and TBP (Mm01277042_m1). Relative changes in mRNA were calculated with the $\Delta\Delta C_t$ method. $C_t$ values of target genes (TG) were calculated relative to a reference gene (RG; TBP or S18) using the following formula $\Delta C_{t_{TG}} = C_{t_{TG}} - C_{t_{RG}}$. Experimental groups (exp) are normalized to control group (con) or in the case of PH with matched resected tissue (res). $\Delta\Delta C_t = \Delta C_{t_{exp}} - \Delta C_{t_{con-or-res}}$. Fold change = $2^{-\Delta\Delta C_t}$. Percentage (%) mRNA remaining = $2^{-\Delta\Delta C_t} * 100$.

## siRNA–liposome preparation and femoral vein injection in C57BL/6 mice

The siRNA sequences MST1 (5′–3′ sense GAGUGUCAAUAUUGC-GAGAtt), MST2 (5′–3′ sense CAAGAGUCAUGAAAAUUGUtt), FVII (Invivofectamine® 2.0 kit) and *in vivo* siRNA negative control (Cat # 4457287) and FVII siRNA (Cat # 4459408) were obtained from Ambion® Life Technologies. Invivofectamine® 2.0 Reagent (Invitrogen) was used for the lipid-based *in vivo* RNAi delivery and siRNA–liposome complexes were prepared according the manufacturer's protocol. Briefly, siRNAs were diluted in DNase, RNase free water to a final concentration of 4 mg/ml. The diluted oligos were mixed 1:1 with the complexation buffer. Invivofectamine was added 1:1 to the siRNA complexation buffer mixture followed by 10-min incubation at room temperature. siRNA–liposome complexes were prepared for injection by dialysis (SPECTRA/Pro). The injected concentration of MST1, MST2 and siRNA negative control siRNA was 1 mg/ml, whereas FVII concentration was 0.3 mg/ml. For *in vivo* delivery, mice were anesthetized by isoflurane inhalation, the femoral vein of the internal side of the hind leg was exposed, and siRNA–liposome complexes were injected 10 μl/g mouse with a 29 G × ½ in needle. After injection, the skin was sutured.

## Genomic microarray

Genomic microarray was performed via the Genomic Platform, University of Geneva. The raw microarray data were background-corrected, normalized using the RMA method as implemented in the R/Bioconductor package affy (Gautier *et al*, 2004). Probe sets were redefined using the alternative chip definition file mogene20stm-mentrezgcdf, as described by Dai *et al* (2005). Differential gene expression was calculated using the moderated *t*-test as described by Ritchie *et al* (2015) and implemented in the R/Bioconductor package limma.

## Pathway analysis

The output of limma was used to perform GSEA using the SetRank method (C. Simillion, R. Liechti, H. Lischer, V. Ioannidis & R. Bruggman, submitted). The key principle of this algorithm is that it discards gene sets that have initially been flagged as significant, if their significance is only due to the overlap with another gene set. It calculates the *P*-value of a gene set using the ranking of its genes in the ordered list of *P*-values as calculated by limma. The following databases were searched for significant gene sets: BIOCYC (Karp *et al*, 2005), Gene Ontology (Ashburner *et al*, 2000), ITFP (Zheng *et al*, 2008), KEGG (Kanehisa *et al*, 2014), LIPID MAPS (Fahy *et al*, 2009), PhosphoSitePlus (Hornbeck *et al*, 2012), REACTOME (Croft *et al*, 2014) and WikiPathways (Kelder *et al*, 2012).

## Measurement of hepatocyte dimensions

The method was adapted from Chen *et al* (2005). Formalin-fixed liver sections were stained for the membrane proteins claudin-3 (Novus Biologicals, Cat. NBP1-35668) and β-catenin (Sigma, Cat. C7082) and visualized by immunofluorescence. Images were acquired with an inverted confocal microscope (Zeiss LSM710).

**The paper explained**

**Problem**
The regenerative capacity of diseased and aged livers is impaired which increases the risk of liver failure following surgical resections. The Hippo pathway controls organ growth by kinase inactivation of the pro-proliferative transcriptional co-activator YAP1.

**Results**
In this study, we demonstrate that proteins in the Hippo pathway are differentially expressed in young vs. aged mice that are dynamically regulated during liver regeneration following partial hepatectomy. By silencing the MST core kinases, we were able to induce pro-proliferative YAP activity and rescue regeneration in an aged mouse model.

**Impact**
Our study supports targeting the Hippo pathway as a means to improve outcomes in regenerative medicine.

The scale was set for each image using the set scale option of ImageJ software. The perpendicular dimensions of cell width and length were annotated by hand using the line tool option. The cell diameter was calculated as the mean of all measured perpendicular dimensions. Four images from randomly selected regions were taken per animal and > 250 measurements were taken per region.

### Measurement of liver injury

Serum samples post-PH or post-injection with siScr or siMST were collected to assess extent of liver injury using assays to measure ALT and AST (P800; Modular Analytics EVO, Roche, Germany).

### Statistics

The graphs and the statistics used in this study have been made by using GraphPad Prism software. *P*-values were calculated using an unpaired, two-tailed Student's *t*-test for all images, except Fig 5C that used a paired, two-tailed Student's *t*-test.

**Expanded View** for this article is available online.

### Acknowledgements
The authors acknowledge the financial support of the Strauss Foundation. We thank the technical contributions of Cynthia Furer, Sarah Overney and Anita Born from Visceral Surgery, University of Bern, and Rémy Bruggmann from the Interfaculty Bioinformatics Unit, University of Bern, for providing computational resources and a special thanks is extended to Dr Erich Cerny for his scientific discussions.

### Author contributions
GL, TM, AK and FB performed the experiments. GL, MM and DS analysed the data. CS performed the bioinformatic analysis. TDH, DC and DS initiated the project. GL and DS wrote the manuscript.

### Conflict of interest
The authors declare that they have no conflict of interest.

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
