## [Review Process File · EMBO Molecular Medicine]

Impaired liver regeneration in aged mice can be rescued by silencing Hippo core kinases MST1 and MST2

Giulio Loforese, Thomas Malinka, Adrian Keogh, Felix Baier, Cedric Simillion, Matteo Montani, Thanos D. Halazonetis, Daniel Candinas and Deborah Stroka

Corresponding author: Deborah Stroka, University of Bern

Review timeline:

Submission date:	24 November 2015
Editorial Decision:	07 January 2016
Revision received:	29 April 2016
Editorial Decision:	16 June 2016
Revision received:	06 October 2016
Editorial Decision:	06 October 2016
Revision received:	24 November 2016
Accepted:	28 October 2016

Transaction Report:

Editor: Roberto Buccione

1st Editorial Decision

07 January 2016

Thank you for the submission of your manuscript to EMBO Molecular Medicine. We have now heard back from the three Reviewers whom we asked to evaluate your manuscript.

Although the Reviewers agree on the potential interest of the manuscript, the issues raised are of a fundamental nature. I will not dwell into much detail, but I would like to highlight the main points.

Reviewer 2 expresses two main and important concerns. The first is that s/he is not satisfied with the quality and depth of analysis of the human sample dataset, including a clear disagreement on the definition of the premenopausal patients; this is not merely a formal issue as there are direct implications on the main conclusions. The other point is that the number of mice used appears insufficient to claim statistical significance. This Reviewer also lists other important items of concern that require your action.

Reviewer 3 notes that the physiological role of PAPP-A is far from proven based on the experimentation. S/he is also concerned that a possible role of progesterone in driving transgene expression has not been excluded. These concerns are of great importance for us as they impinge on the most interesting potential messages of the manuscript. Reviewer 3 also laments issues with statistics and numbers of experimental animals, and lists a number of other relevant points.

Reviewer 1 is less reserved but also raises the issue of the physiological role of PAPP-A. S/he also provides an extensive list of items that need your action, which all appear important but feasible and

should be addressed in full.

In conclusion, while publication of the paper cannot be considered at this stage, given the potential interest of your findings and after internal discussion, we have decided to give you the opportunity to address the criticisms. We are thus prepared to consider a substantially revised submission, with the understanding that the Reviewers' concerns must be addressed with additional experimental data where appropriate and that acceptance of the manuscript will entail a second round of review. The overall aim is to significantly upgrade the relevance and conclusiveness of the dataset, which of course is of paramount importance for our title.

Please note that it is EMBO Molecular Medicine policy to allow a single round of revision only and that, therefore, acceptance or rejection of the manuscript will depend on the completeness of your responses included in the next, final version of the manuscript.

EMBO Molecular Medicine now requires a complete author checklist (<http://embomolmed.embopress.org/authorguide#editorial3>) to be submitted with all revised manuscripts. Provision of the author checklist is mandatory at revision stage; The checklist is designed to enhance and standardize reporting of key information in research papers and to support reanalysis and repetition of experiments by the community. The list covers key information for figure panels and captions and focuses on statistics, the reporting of reagents, animal models and human subject-derived data, as well as guidance to optimise data accessibility. This checklist especially relevant in this case given the issues raised with respect to statistical treatment and animal numbers.

As you know, EMBO Molecular Medicine has a "scooping protection" policy, whereby similar findings that are published by others during review or revision are not a criterion for rejection. However, I do ask you to get in touch with us after three months if you have not completed your revision, to update us on the status. Please also contact us as soon as possible if similar work is published elsewhere.

I also suggest that you carefully adhere to our guidelines for publication in your next version, including our new requirements for supplemental data (see also below) to speed up the pre-acceptance process in case of a positive outcome.

I look forward to seeing a revised form of your manuscript as soon as possible.

***** Reviewer's comments *****

Referee #1 (Remarks):

In their manuscript "Rescue of impaired liver regeneration in aged mice by silencing the Hippo pathway", Loforese et al. show that therapeutic targeting of the hippo pathway can improve liver regeneration after partial hepatectomy in aged mice. Facilitating regeneration, especially in the liver is of great interest, as chronic or acute liver failure represent a major global health issue.

MST1/2 inhibition as a potential therapy to rescue impaired liver regeneration is in accordance with current literature. It has previously been shown that YAP and MST1/2 play a role in hepatic proliferation and organ overgrowth.

Nevertheless, the authors clearly go beyond published data, as they apply advanced technology (liposome encapsulated siRNAs) *in vivo* to functionally show that liver regeneration in old mice can be rescued by *in vivo* delivery of siRNAs silencing MST1/2.

There are a few points that need attention before publication can be recommended:

*LATS-1 detection via western blot. In the manuscript a band of 130 kDa is described which is not labeled in the blot in figure 1B nor 3B whereas a band at 120 kDa is highlighted. In the manufacturers description of the used abcam-antibody the following statement can be found: "Ab70651: Detects a band of approximately 160 kDa (predicted molecular weight: 127 kDa)." Unfortunately the blot to identify

the right pLATS1 band is not in accordance with the blot shown in figure 1B. In S1 only one band for pLATS1 is detected at 24 h and no band at 0 h. In Figure 1B for 0h a lower band is detected and for 24 h two bands are shown. This raises questions regarding the reproducibility of these blots and the correct identification of the protein. A knockdown experiment for LATS1 could help to clarify this issue.

* The authors should elucidate whether MST1 and MST2 are equally important for liver regeneration. So far only phosphorylation of MST1 is shown. In my eyes this is not only interesting from a mechanistic point of view but also for a potential translation of the results.

Referee #2 (Comments on Novelty/Model System):

It was shown previously that loss of MST function results in hepatocyte proliferation. It is no surprise that this also happens in old mice. Nevertheless, this may be relevant for regenerative medicine of course.

Referee #2 (Remarks):

In the manuscript of Loforese et al, the authors claim that the Hippo pathway effector YAP is activated during the proliferation phase of regeneration after partial hepatectomy. Moreover, the authors propose that downregulation of the Hippo pathway kinases MST1 and MST2 restores regeneration in aged livers by hyperactivation of YAP. Altogether, they conclude that silencing the Hippo pathway kinases MST1 and MST2 can be used as a therapeutic approach to improve regeneration.

Although inducing regeneration of non-regenerating organs has broad therapeutic potential, and thus is of great interest, this manuscript is not supported by enough data and there is little mechanistic insight. Several issues need further examination.

- Although it is well known that YAP is highly enriched in cholangiocytes and not in hepatocytes, all analysis of RNA and protein levels are done from total liver extracts. Thus, this clearly complicates the interpretation of the data as it is difficult to know if the changes in protein or gene expression levels happened in hepatocytes or other cell types. The increase in YAP levels might simply reflect growth of the biliary tree and not hepatocyte proliferation. Thus the paper would benefit much if the analysis could be repeated on purified hepatocytes (or other cell types).

- The authors find that MST and LATS are activated early during regeneration and they also mention that they do not see a decrease of either below normal levels at later time points, during the proliferative phase of regeneration. How does this fit together with the claim that YAP is activated during regeneration? This makes no sense. Either, YAP is activated by mechanisms other than reduction of phosphorylation and/or the proteins are expressed in different cell types (see above). Clearly, the presented analysis is premature to draw conclusions.

- Fig 2: I do not see significant differences in some of the samples. All Westerns need to be quantified and statistically analyzed. Also, In general, the data shown in Western blots is difficult to interpret as most of them lack of loading controls.

- Also, does the upregulation of MST result in a concordant increase in pYAP as would be expected?

- The authors show that YAP levels are increased during regeneration, but is it actually required for regeneration? Without characterizing the function of YAP during liver regeneration, the analysis of YAP levels and its target genes is more circumstantial rather than conclusive.

- More importantly, what happens to YAP in old mice after partial hepatectomy?

- The effect of MST1 KD seems to be very strong on MST1 protein levels, yet not so much in phospho LATS and phospho YAP (Figure 3). Not clear if MST1 and MST2 were simultaneously KD.

- Although the authors propose that downregulation of MST could be used as a therapeutic approach, surprisingly the authors do not demonstrate how normal or abnormal the MST KD regenerated livers are. There is no analysis of what happens to the different cell types in the liver (marker expression), and there is no liver function analysis shown for the MST KD aged livers after PHx.

Referee #3 (Remarks):

In this work Loforese and colleagues first evaluate the regulation of the Hippo pathway during liver regeneration after partial hepatectomy (PH) in mice. The authors cogently demonstrate the activation of the core kinases in this pathway, MST1 and LATS1, early after tissue resection, before the onset of the proliferative phase, coincidentally with the activation of YAP1 target genes and subsequent hepatocellular proliferation. Concomitantly, they show that in aged mice the Hippo pathway is anomalous in livers that displayed an impaired regenerative response, which is frequently accompanied by the death of hepatectomized aged mice. In view of these original findings the authors hypothesize that pharmacological intervention aimed at the inhibition of Hippo pathway could be a means to improve liver regeneration in old animals. To this end they devise effective RNAi tools to inhibit MST1/2 gene expression in mouse liver and test their efficacy. Interestingly, inhibition of MST1/2 expression in aged mice had a remarkable effect on the survival and regenerative response of these animals after PH. This is a very nice study, well designed, and presenting sound data that are very well discussed. The observations reported here can have translational/clinical implications in the field of acute liver injury and regeneration. The development of hepatoprotective and liver pro-regenerative strategies is still an unmet medical need, and this study really advances in that direction. I only have some aspects that if properly addressed can increase the significance of this otherwise excellent work.

1) The authors demonstrate the regulation of MST1 and LATS1 during the phases of liver regeneration (Figure 1B) and go on to compare the difference of this regulation in young versus aged mice (Figure 2C). However they do not provide data on LATS. Is LATS1 phosphorylation also altered in aged mice during liver regeneration? What is the status of Yap phosphorylation and subcellular localization?

2) In non-regenerating aged animals Yap target genes are used as a read-out for Yap activity and hepatocyte proliferation. Can they provide additional markers to validate the liver's regenerative status, for example expression of the cell cycle proteins, Cyclin A2, D and Cdc25B, and expression of the cell cycle inhibitors, i.e. p21.

3) Can the authors clarify the discrepancy in which they consider the < 10% Ki67 positive hepatocytes as non-regenerating in Fig 2A, and 6-8% Ki67 positive hepatocytes as regenerating in Fig 3E?

1st Revision - authors' response

29 April 2016

Referee #1

In their manuscript "Rescue of impaired liver regeneration in aged mice by silencing the Hippo pathway", Loforese et al. show that therapeutic targeting of the hippo pathway can improve liver regeneration after partial hepatectomy in aged mice. Facilitating regeneration, especially in the liver is of great interest, as chronic or acute liver failure represents a major global health issue.

MST1/2 inhibition as a potential therapy to rescue impaired liver regeneration is in accordance with current literature. It has previously been shown that YAP and MST1/2 play a role in hepatic proliferation and organ overgrowth. Nevertheless, the authors clearly go beyond published

data, as they apply advanced technology (liposome encapsulated siRNAs) in vivo to functionally show that liver regeneration in old mice can be rescued by in vivo delivery of siRNAs silencing MST1/2.

There are a few points that need attention before publication can be recommended:

> LATS-1 detection via western blot. In the manuscript a band of 130 kDa is described which is not labeled in the blot in figure 1B nor 3B whereas a band at 120 kDa is highlighted. In the manufacturers description of the used Abcam-antibody the following statement can be found: "Ab70651: Detects a band of approximately 160 kDa (predicted molecular weight: 127 kDa)." Unfortunately the blot to identify the right pLATS1 band is not in accordance with the blot shown in figure 1B. In S1 only one band for pLATS1 is detected at 24 h and no band at 0 h. In Figure 1B for 0h a lower band is detected and for 24 h two bands are shown. This raises questions regarding the reproducibility of these blots and the correct identification of the protein. A knockdown experiment for LATS1 could help to clarify this issue.

>Reply: We thank the reviewer for pointing out the discrepancy with our LATS protein data. We have corrected the molecular weight in the manuscript to our observed molecular weight of approximately 150kDa (Fig1B & 3B). Antibodies from Abcam and Cell Signalling were used to identify LATS and p-LATS, respectively. Both companies report a detectable double band between 130 and 170 kDa (there is a bit of discrepancy).

Working directly with the technical service department at Cell Signaling, we used their positive control (HeLa cells +/-TPA) to identify LATS as the slower migrating band at approximately 150 kDa. We increased the exposure time while developing the Western to detect both bands in our liver extracts in the control experiment for comparison. For this reason the HeLa cell signal is overexposed (Appendix Figure S2). In addition, and for certainty, we knocked down LATS in the Hep3B liver cell line with siRNA. In 10µg of Hep3B cell extracts only the upper 150kDa band was detectable which was diminished with siRNA targeting LATS (Appendix Figure S2).

> The authors should elucidate whether MST1 and MST2 are equally important for liver regeneration. So far only phosphorylation of MST1 is shown. In my eyes this is not only interesting from a mechanistic point of view but also for a potential translation of the results.

>Reply: The reviewer brings forward a very interesting point. In the liver, MST1 and MST2 kinases were described to have redundant function and knock out of either MST1 or 2 does not result in the over growth of the liver ¹. Until now individual/unique functions of MST1 and 2 in the Hippo pathway are not well known. However, MST1 functions in other pathways, such as negatively regulating AKT and mTOR activity ². At this point, we cannot rule out that knock down of MST1 and MST2 effects pathways in addition to Hippo pathways. Further characterization of MST2 function in both Hippo and non-Hippo signaling pathways also need to be done (very interesting, but out of the scope of this study). To explore their importance in our model is technically challenging, as we do not expect a phenotype with a single knockdown. Moreover, there is no antibody available that recognizes phosphorylated MST2 specifically. The antibody we used recognizes phospho-MST1 and MST2 at Thr183 and 180, respectively. Nevertheless, to add to what we know about the regulation of MST2 in liver regeneration we can demonstrate that MST2 mRNA is not regulated during liver regeneration (Appendix Figure S3). We performed a Western blot for MST2 and have included it in the supplementary data (Appendix Figure S1), however, due to the quality of the antibody, we hope that the reviewer can accept that we do not wish to draw any conclusions regarding the regulation of MST2 protein in mouse liver tissue.

Referee #2

It was shown previously that loss of MST function results in hepatocyte proliferation. It is no surprise that this also happens in old mice. Nevertheless, this may be relevant for regenerative medicine of course.

> In the manuscript of Loforese et al, the authors claim that the Hippo pathway effector YAP is activated during the proliferation phase of regeneration after partial hepatectomy. Moreover, the authors propose that downregulation of the Hippo pathway kinases MST1 and MST2 restores regeneration in aged livers by hyperactivation of YAP. Altogether, they conclude that silencing the Hippo pathway kinases MST1 and MST2 can be used as a therapeutic approach to improve regeneration. Although inducing regeneration of non-regenerating organs has broad therapeutic potential, and thus is of great interest, this manuscript is not supported by enough data and there is little mechanistic insight. Several issues need further examination.

>Reply: We thank the reviewer for their constructive remarks. Addressing these concerns has improved the manuscript considerably.

> Although it is well known that YAP is highly enriched in cholangiocytes and not in hepatocytes, all analysis of RNA and protein levels are done from total liver extracts. Thus, this clearly complicates the interpretation of the data as it is difficult to know if the changes in protein or gene expression levels happened in hepatocytes or other cell types. The increase in YAP levels might simply reflect growth of the biliary tree and not hepatocyte proliferation. Thus the paper would benefit much if the analysis could be repeated on purified hepatocytes (or other cell types).

>Reply: In order to address the reviewers concerns and to demonstrate that YAP protein is present in hepatocytes as well as in cholangiocytes, we purified and cultured both cell populations from normal liver tissue. By Western blot, we are able to demonstrate high levels of YAP and TAZ protein in cultured hepatocytes and cholangiocytes (This data has been added as an expanded view data, Figure EV1). In addition, we isolated hepatocytes from sham operated and regenerating mouse livers (48 hours post PH) and demonstrate by Western blot that YAP and TAZ are highly increased in hepatocytes isolated from regenerating mouse livers. This figure has been added to the manuscript as Figure 1E. Taken together, these experiments provide evidence that YAP levels are most likely increased in the parenchymal cells during liver regeneration. Moreover, with our control siRNA experiment targeting FVII, we can be certain that our siRNA did in fact reach the hepatocyte population to decrease MST expression levels (Figure EV2).

> The authors find that MST and LATS are activated early during regeneration and they also mention that they do not see a decrease of either below normal levels at later time points, during the proliferative phase of regeneration. How does this fit together with the claim that YAP is activated during regeneration? This makes no sense. Either, YAP is activated by mechanisms other than reduction of phosphorylation and/or the proteins are expressed in different cell types (see above). Clearly, the presented analysis is premature to draw conclusions.

>Reply: The reviewer raises an interesting observation in our study. Although the increase of MST and LATS during the early phases of liver regeneration is contrary to what one may predict, we found this observation interesting and worthy to report. However, we carefully have not drawn strong conclusions at this point and correlate the increased activity of the kinases to the hypertrophic phase of the regenerative response. This phase is before the proliferative/YAP active phase. We also suggested in the discussion that the flux in protein levels may be a regulating factor in addition to their absolute protein levels. Moreover, one can hypothesize that the steady state levels of pMST and pLATS needed to maintain YAP in a quiescent liver are not sufficient to control its expression when the regenerative signals are fully engaged, as occurring following PH (text added to discussion, page 18).

> Fig 2: I do not see significant differences in some of the samples. All Westerns need to be quantified and statistically analyzed. Also, In general, the data shown in Western blots is difficult to interpret as most of them lack of loading controls.

> Reply: We have added loading controls to all Western blots and have quantified the blots that needed better clarification for the interpretation of the results (particularly, Figure 1B, and 2C).

> The authors show that YAP levels are increased during regeneration, but is it actually required for regeneration? Without characterizing the function of YAP during liver regeneration, the analysis of YAP levels and its target genes is more circumstantial rather than conclusive.

> Reply: The reviewer raises a very intriguing question and is correct with regards to questioning whether YAP levels and its target gene expression are actually required for liver regeneration or are merely circumstantial. There is evidence in the literature in which loss of YAP expression³ or interfering with YAP activity⁴ impairs hepatocyte proliferation, however we are unaware if PH studies have actually been performed in liver specific YAP or YAP/TAZ knock out mice. Zhang et al⁵, reported that Alb-Cre;Yap^{fllox/fllox} mice (with deletion of YAP at perinatal stage E18 to P1) were born with expected Mendelian ratio and no overt abnormalities but developed an abnormal biliary system and defects in hepatocyte survival. They reported, interestingly, an increased turnover of hepatocytes in Yap-deficient livers. It is well accepted that there is a high redundancy of signaling pathways that are activated during liver regeneration following PH. In this regard, YAP may function to promote hepatocyte survival and to foster cell proliferation. We can postulate that loss of YAP will adversely affect regeneration, by compromising hepatocyte viability and thereby leading to a delay in the regenerative process. Nevertheless, YAP activation can be used as a marker of a regenerating liver and when its expression and/or function is altered gives an indication of the regenerative status of the liver. The future goal of our research would be to compare regeneration following PH in Alb-Cre;Yap^{fllox/fllox} or Alb-Cre;TAZ/Yap^{fllox/fllox} mice, however, we suspect others are also working on this topic and this data may soon be available from other groups.

> More importantly, what happens to YAP in old mice after partial hepatectomy?

>Reply: We have added data for YAP and TAZ protein to Figure 2 comparing their expression in young versus aged mice. YAP protein is increased in young animals following PH, whereas in aged animals there is a higher steady state level that is not increased in aged regenerating livers (new Figure 3A). This observation is unique to YAP, as TAZ expression is increased in the same manner in both young and aged mice (Figure 3A). Taken together, this additional data supports that the Hippo pathway is impaired in non-regenerating aged livers.

> The effect of MST1 KD seems to be very strong on MST1 protein levels, yet not so much in phospho LATS and phospho YAP (Figure 3). Not clear if MST1 and MST2 were simultaneously KD.

>Reply: We demonstrate in Figure 4A and Figure EV5, the effect of KD on MST1 and MST2 mRNA expression. Both mRNAs were significantly lower than the scrambled controls, albeit the effect on MST2 was not as strong. Yes- we agree the effect of MST KD has a subtle effect of p-LATS and p-YAP that required time (day 6) to become evident, reflecting the overall subtle effect we observe on liver cell proliferation shown as an increase of Ki67 positive hepatocytes in MST KD livers compared to the scrambled control. With the available antibody detecting MST2 protein, we could not obtain a satisfactory result to report MST2 levels in mouse liver tissue (very high background).

> Although the authors propose that down regulation of MST could be used as a therapeutic approach, surprisingly the authors do not demonstrate how normal or abnormal the MST KD regenerated livers are. There is no analysis of what happens to the different cell types in the liver (marker expression), and there is no liver function analysis shown for the MST KD aged livers after PHx.

> Reply: In the supplementary data (Appendix Figure S6) we report that there was no increase of serum transaminases in the MST KD livers indicating no liver injury in young animals. In addition, we have measured liver transaminase (ALT and AST) and alkaline phosphatase (ALP) in the regenerating siMST aged animals compared to aged only (Appendix Figure S8). Serum ALT, AST and ALP levels were elevated following PH, indicating liver cell injury in aged animals, however, we report no significant differences in regenerating aged mice treated with siMST compared to regenerating aged liver controls. In addition, we calculated the hepatocyte geometric diameter as an indication of the hypertrophic state of hepatocytes to determine if the increase of liver to body/weight ratio was due to an increase of hepatocyte size and/or number (new data in Figure 4F). Finally, we stained liver sections with CK19 to monitor and quantitate bile duct proliferation. In our model, there were no significant differences in bile duct number between control and siMST KD groups. As the duration of our experiment is too short, and we

did not target Nf2 in order to provoke a duct reaction, we have therefore have not included this data in the manuscript.

Referee #3

In this work Loforese and colleagues first evaluate the regulation of the Hippo pathway during liver regeneration after partial hepatectomy (PH) in mice. The authors cogently demonstrate the activation of the core kinases in this pathway, MST1 and LATS1, early after tissue resection, before the onset of the proliferative phase, coincidentally with the activation of YAP1 target genes and subsequent hepatocellular proliferation. Concomitantly, they show that in aged mice the Hippo pathway is anomalous in livers that displayed an impaired regenerative response, which is frequently accompanied by the death of hepatectomized aged mice. In view of these original findings the authors hypothesize that pharmacological intervention aimed at the inhibition of Hippo pathway could be a means to improve liver regeneration in old animals. To this end they devise effective RNAi tools to inhibit MST1/2 gene expression in mouse liver and test their efficacy. Interestingly, inhibition of MST1/2 expression in aged mice had a remarkable effect on the survival and regenerative response of these animals after PH. This is a very nice study, well designed, and presenting sound data that are very well discussed. The observations reported here can have translational/clinical implications in the field of acute liver injury and regeneration. The development of hepatoprotective and liver pro-regenerative strategies is still an unmet medical need, and this study really advances in that direction. I only have some aspects that if properly addressed can increase the significance of this otherwise excellent work.

> The authors demonstrate the regulation of MST1 and LATS1 during the phases of liver regeneration (Figure 1B) and go on to compare the difference of this regulation in young versus aged mice (Figure 2C). However they do not provide data on LATS. Is LATS1 phosphorylation also altered in aged mice during liver regeneration?

> Reply: In young mice we observe an increase of LATS phosphorylation 6-24 h following PH, whereas in aged animals, steady-state p-LATS levels were increased with no increased phosphorylation following PH (Appendix Figure S4). In addition, YAP protein is increased in young animals following PH, whereas, in aged animals there is a higher steady state level that is not increased in aged regenerating livers (new Figure 3A). We observe a loss of YAP phosphorylation in young mice following PH, whereas no change is observed in aged animals (new Figure 3A). This observation is unique to YAP, as TAZ is regulated in the same manner in both young and aged mice (new Figure 3A). Taken together, this additional data support that the Hippo pathway is impaired in non-regenerating aged livers and that aged mice can be used as a model of impaired liver regeneration.

> In non-regenerating aged animals Yap target genes are used as a read-out for Yap activity and hepatocyte proliferation. Can they provide additional markers to validate the liver's regenerative status, for example expression of the cell cycle proteins, Cyclin A2, Cyclin D and Cdc25B, and expression of the cell cycle inhibitors, i.e. p21.

> Reply: To provide further evidence of the liver's regenerative status, we have included additional markers to validate liver regeneration, namely, cyclin A2, cyclin D1 and the cell cycle inhibitor p21 (Figure 3B).

> Can the authors clarify the discrepancy in which they consider the < 10% Ki67 positive hepatocytes as non-regenerating in Fig 2A, and 6-8% Ki67 positive hepatocytes as regenerating in Fig 3E?

> Reply: In Fig 2A, we are looking at the percentage of Ki67+ cells in response to partial hepatectomy, a well-established model to stimulate hepatocyte proliferation. In healthy young mice, the percentage of positive hepatocytes at 40 h post PH can reach 60-70%, therefore a proliferation index of 10% or lower indicates a liver that is not regeneration/regeneration is impaired. This is in contrast to the condition in which we aim to provoke proliferation in a

quiescent liver (Fig 2A). Here we demonstrate that only targeting MST kinases is enough to provoke 6-8% of hepatocytes to proliferate in the quiescent liver. A study published while our manuscript was under review has demonstrated that combined targeting of MST1, MST2 and Nf2 was necessary to produce a signal strong enough to provoke liver overgrowth after 15 days³. This paper has now been referred to in the discussion.

References

- 1 Zhou, D. *et al.* Mst1 and Mst2 maintain hepatocyte quiescence and suppress hepatocellular carcinoma development through inactivation of the Yap1 oncogene. *Cancer cell* **16**, 425-438 (2009).
- 2 Chao, Y. *et al.* Mst1 regulates glioma cell proliferation via the AKT/mTOR signaling pathway. *Journal of neuro-oncology* **121**, 279-288 (2015).
- 3 Yin, H. RNAi-nanoparticulate manipulation of gene expression as a new functional genomics tool in the liver. *Journal of hepatology* (2016).
- 4 Liu-Chittenden, Y. *et al.* Genetic and pharmacological disruption of the TEAD-YAP complex suppresses the oncogenic activity of YAP. *Genes & development* **26**, 1300-1305 (2012).
- 5 Zhang, N. *et al.* The Merlin/NF2 tumor suppressor functions through the YAP oncoprotein to regulate tissue homeostasis in mammals. *Developmental cell* **19**, 27-38 (2010).

2nd Editorial Decision

16 June 2016

Thank you for the submission of your revised manuscript to EMBO Molecular Medicine. We have now heard back from the two Reviewers whom we asked to evaluate your manuscript.

I apologise for significant delay in getting back to you. We experienced difficulties in obtaining the reviewer evaluations in a timely manner. In addition to this, your manuscript required further in depth discussion with my colleagues and the reviewers on the way forward.

You will see that, while reviewer 1 (pending a clarification) and reviewer 3 are mostly satisfied that their concerns have been adequately addressed, reviewer 2 remains quite reserved and feels that important mechanistic conclusions have not been adequately supported. These include direct evidence of the requirement for YAP in liver regeneration, discrepancies between LATS1/2 and YAP/TAZ activities, and the fact that cell type analysis is missing for the MST knockdown regenerated livers. I acknowledge that these issues were brought up in the first round of evaluations.

After internal discussion and additional cross commenting with the reviewer who accepted to do so, we agreed that the message of your study is compelling and has translational value. We also agreed that basic evidence on the role of YAP in hepatocellular growth and proliferation is already available, and that this manuscript does demonstrate the feasibility of targeting this pathway in in specific clinical settings.

Although we would normally not allow a second extensive revision, based on our discussions I am prepared in this case however, to give you the opportunity to improve your manuscript. Although I do encourage you to develop the study as far as realistically possible in a mechanistic sense as indicated by the reviewer, I would especially ask you to make the effort to carry out a cell type (marker) analysis on the MST KD livers as requested. In any case please provide a point-by-point rebuttal on Reviewer 2's comments. Please also comply with Reviewer 1's request on figure EV1.

As you know, EMBO Molecular Medicine has a "scooping protection" policy, whereby similar findings that are published by others during review or revision are not a criterion for rejection. However, I do ask you to get in touch with us after three months if you have not completed your revision, to update us on the status. Please also contact us as soon as possible if similar work is published elsewhere.

I also suggest that you carefully adhere to our guidelines for publication in your next version,

including our new requirements for supplemental data (see also below) to speed up the pre-acceptance process in case of a positive outcome. There are still a few issues with the format that need to be fixed prior to eventual publication. Do not hesitate to contact our editorial office for further assistance. We also note that the "The Paper Explained" section is still incomplete. Please refer to any of our published articles as an example..

I look forward to seeing a revised form of your manuscript as soon as possible.

***** Reviewer's comments *****

Referee #1 (Remarks):

The authors have addressed the critique of all referees in a sufficient way. Just one minor comment concerning the quality of Figure EV1. In this figure the role of YAP and TAZ should be shown in hepatocytes versus cholangiocytes. The brightfield pictures of these cells look convincing but a simple addition of HNF4alpha and CK19 westernblots as specific markers for both populations could easily confirm the purity.

Referee #2 (Remarks):

Although the authors replied to the reviewer's questions, they mainly did so by discussing existing literature and not by providing new experimental evidence. Thus, in the absence of experimental proof, many critical questions remained unsatisfactorily answered.

One of the most important questions the authors were asked was to test if the increase in YAP is required for regeneration or if it is just circumstantial. However, despite the importance of this question, it was not answered experimentally. In the absence of YAP and TAZ mutant mice, the authors could have use the same method they used for MST1/2 KD, namely to do siRNA to downregulate YAP and TAZ and test its effect after PHx. With the presented data it is not possible to conclude whether YAP is actually required for regeneration after PHx, and even less to conclude that YAP is actually regulated by MST/LATS in aged mice after PHx.

In Figure S4 they show that pLATS levels don't increase at 40h after PHx (despite the increase in pMST shown in fig3A). The authors also show that the levels of YAP and pYAP don't change between young and old at 40h after PHx, and the levels of TAZ increase in aged livers after PHx (figure 3A). Thus, if LATS1/2 are not more phosphorylated, the levels of YAP don't change and the levels of TAZ increase, why should YAP and TAZ not be fully active in old mice? And, why increased levels of TAZ are not sufficient to induce proliferation after PHx? Again, are YAP and TAZ even required? What happens when YAP/TAZ are KD in MST KD hepatocytes? Can this suppress the MST KD effect? If MST KD has only subtle effects on pLATS and pYAP, and a subtle proliferation effect, then what is the role of MST? Answering these questions is important because if MST KD doesn't affect the activity of YAP/TAZ, then there is no molecular mechanism/effector to explain the effect they see in old mice treated with the MST siRNAs.

Also, although now they included ALT and ALS serum analysis, they did not show how normal or abnormal the MST KD regenerated livers are in terms of cells types (marker analysis). What happens to ductal cells? Given that in all models of YAP gain of function there is oval cell expansion, are there oval cells arising in the MST KD after PHx?

Minor remarks:

In Figure S7, it seems that all the small hepatocytes were excluded from the quantification. Why? In Fig 5B only KI67 staining for aged mice with MST KD + PHx is shown, but not for the non-targeting KD control. How can we tell that there is any difference in KI67?

Referee #3 (Comments on Novelty/Model System):

The study is well designed and performed, and the information provided is novel in the field, and

with potential translational implications in the context of aged liver regeneration. This study is highly suitable for publication in Embo Molecular Medicine

Referee #3 (Remarks):

The authors properly addressed all my concerns.

2nd Revision - authors' response

06 October 2016

The major changes and additions to the manuscript are highlighted in yellow in the revised version of the text.

***** Reviewer's comments *****

Reviewer #1

The authors have addressed the critique of all referees in a sufficient way. Just one minor comment concerning the quality of Figure EV1. In this figure the role of YAP and TAZ should be shown in hepatocytes versus cholangiocytes. The bright field pictures of these cells look convincing but a simple addition of HNF4alpha and CK19 western blots as specific markers for both populations could easily confirm the purity.

Response: We thank the reviewer for their positive approval of our first response letter. As requested, we have added a western blot for HNF4 α and CK-19 in Figure EV1 to demonstrate the purity of our isolated cell populations. In addition, we also added to EV1 a photomicrograph of a mouse liver immunostained with the antibody used to detect HNF4 α by Western to show its specificity for hepatocytes.

Reviewer #2

This reviewer raises several important questions and we have attempted to answer them adequately within the scope of this study. I have divided their comments into 6 parts in order to address them by topic.

1) *Although the authors replied to the reviewer's questions, they mainly did so by discussing existing literature and not by providing new experimental evidence. Thus, in the absence of experimental proof, many critical questions remained unsatisfactorily answered.*

I would like to kindly point out that many points from the first review were addressed with new data and this data was added to the revised manuscript:

1. To address the specificity of Yap staining in hepatocytes versus cholangiocytes. Figure 1E and Figure EV1
2. Quantification of all Westerns: Quantifications were added to Figure 1B and 1C.
3. Yap expression in aged mice following PH: Figure 3A western
4. Liver injury in aged animals following MST KD: Figure S8: ALT, AST and ALK measurements and Figure 4F: hepatocyte size
5. KD of LATS and Western to show antibody specificity: Figure S2
6. Expression levels of MST2 (mRNA and protein): Figure S1 and S3
7. Status of LATS following PH in young and aged mice: Figure S4
8. RT-qPCR for cell cycle genes: Figure 3B

2) *One of the most important questions the authors were asked was to test if the increase in YAP is required for regeneration or if it is just circumstantial. However, despite the importance of this question, it was not answered experimentally. In the absence of YAP and TAZ mutant mice, the authors could have used the same method they used for MST1/2 KD, namely to do siRNA to downregulate YAP and TAZ and test its effect after PHx. With the presented data it is not possible to conclude whether YAP is actually required for regeneration after PHx, and even less to conclude that YAP is actually regulated by MST/LATS in aged mice after PHx.*

The data present in the community and made available to us by one of our collaborators demonstrates that livers of Yap KO mice do regenerate less. However, this is misleading, as the

livers of Yap KO mice do not develop normally and therefore are histologically abnormal and therefore one cannot differentiate between the impairment of regeneration as a result of loss of Yap or as a result of the starting atypical liver. Although it shows less regeneration, we and others, in good scientific practice, would not use the Yap KO mice to demonstrate the requirement of YAP in liver regeneration. We have now initiated a collaboration to generate a conditional Yap and Taz KO in the liver, with which we can assess the activity of Yap/Taz in adult liver tissue. It is hopeful that these mice will be able to address the question of whether Yap and/or Taz are required for liver regeneration, however, these mice will be the basis for a future studies and this clearly cannot be concluded at in a timely manner to be included within this manuscript. The reviewer also suggested that we should knock down Yap with siRNA. We considered this long and carefully and decided it would be best to invest our resources to have the conditional KO animals, as the siRNA experiments would not be as robust. In the future, these animals will be valuable to address the mechanistic questions raised by the reviewer, particularly when targeting upstream targets. Although we used siRNA methodology to prove the feasibility of targeting the Hippo pathway to improve liver regeneration, we also learned of the limitation of the methodology, namely the toxicity of the liposome delivery as an unnecessary negative side effect. Therefore we would invest time and resources in the wrong direction without the full confidence that it was the right approach to address this question.

Coming back to the reviewers questions of whether Yap is **actually required** for regeneration and is it **actually regulated by MST/LATS** after PH...

... our data and others show that YAP is active in regenerating livers. However, we cannot tell at this time the individual contribution of YAP and TAZ. It is well accepted that there is a high redundancy of signaling pathways that are activated during liver regeneration following PH. In this regard, we can postulate that loss of YAP will adversely affect regeneration, by compromising hepatocyte viability and thereby leading to a delay in the regenerative process. Nevertheless, we are using YAP activation as a marker of a regenerating liver and when its expression and/or function is altered gives an indication of the regenerative status of the liver. The more difficult question is to know the mechanism of how and if YAP/TAZ are regulated by MST and LATS after PH. We show the expression status of the key players of the pathway, however, we are cautious in our interpretation of the results as to assume a linear and simple association between them.

Moreover, in the time that our manuscript has been under review, a study has been published in Science Translation Medicine (Fan, et al., 17. August 2016) demonstrating that pharmacological targeting of MST1 and MST2 augments liver regeneration. In this study, they use Yap activation and biological readouts of liver regeneration and repair to show the efficacy of their compound XMU-MP-1.

3) If LATS1/2 are not more phosphorylated, the levels of YAP don't change and the levels of TAZ increase, why should YAP and TAZ not be fully active in old mice? And, why increased levels of TAZ are not sufficient to induce proliferation after PHx? Again, are YAP and TAZ even required? What happens when YAP/TAZ are KD in MST KD hepatocytes? Can this suppress the MST KD effect? If MST KD has only subtle effects on pLATS and pYAP, and a subtle proliferation effect, then what is the role of MST? Answering these questions is important because if MST KD doesn't affect the activity of YAP/TAZ, then there is no molecular mechanism/effector to explain the effect they see in old mice treated with the MST siRNAs.

The reviewer raises several interesting questions based on our experimental observations that we cannot answer at this time. We cannot answer why increased levels of TAZ are not sufficient to induce proliferation after PHx or the observed levels of LATS protein. Nevertheless, we are reporting our observation of the regulation of these proteins in a robust model of liver regeneration as we obtained them.

MST KD does affect the activity of YAP/TAZ, as the liver's are regenerating and YAP target genes are active, however, it is uncertain if the effect is direct or via another intermediate. At this point we are investigating how inhibition of MST improves liver regeneration and the trying to uncover the mechanism of its function.

4) Also, although now they included ALT and ALS serum analysis, they did not show how normal or abnormal the MST KD regenerated livers are in terms of cells types (marker analysis). What

happens to ductal cells? Given that in all models of YAP gain of function there is oval cell expansion, are there oval cells arising in the MST KD after PHx?

We focused on the ductal cells/ductal reactions in order to address the question of how normal or abnormal the MST KD regenerating livers are. For comparison we used young regenerating livers at the same time point post PH. To ensure that our assessment was correct and justly representing our data we asked for the collaboration of a liver pathologist working at the Institute of Pathology at the University of Bern. Based on his assessment of the liver tissues and as depicted in Figures 2A and 5C, we conclude that there are no abnormalities between control young regeneration livers and aged liver with MST KD. This conclusion is based on no increase of bile ducts or signs of ductal reactions, which we visualized, by hematoxylin and eosin staining, Ki67 and CK-19 staining. In addition, there were no signs of oval cell expansion.

As an explanation why we do not see a ductal reaction, the Yap ‘gain of function’ transgenic models are much stronger and longer in duration, thereby resulting in a ductal/oval cell response. We believe that our model, which is a transient KD and over a shorter time span of maximum 64 h post KD (and 40 h post PH) is too mild/short to achieve this predicted response. Once again referring the new manuscript mentioned above by Fan et al., they report “very low levels of oval cell expansion” in mice treated for 2 months with an MST inhibitor XMU-MP-1 (page 5, right column and Fig 4D), thus further supporting our results after 2 days.

Minor Remarks:

5) In Figure S7, it seems that all the small hepatocytes were excluded from the quantification. Why?

Response: We rechecked and confirmed the criteria used for our analysis and we do not feel that small hepatocytes were excluded from the quantification. All hepatocytes with a visible membrane staining in which an accurate measurement could be made were included. Plot of the size distribution of the cells counted are provided below, which is a reflection of data used in the box plot for Figure 5G.

6) In Fig 5B, only Ki67 staining for aged mice with MST KD + PHx is shown, but not for the non-targeting KD control. How can we tell that there is any difference in KI67?

Response: The non-targeting KD control animals died within the first 24 hours following PH and we clearly stated we left them out of further analysis, so we were not reproached for performing analysis on tissues from non-viable mice. Also, as we show in Figure 1A, we do not expect any Ki67 positive cells 24 h post PH. Nevertheless, as we harvested the livers from some of these animals we performed a Ki67 staining and there were < 1% Ki67 positive cells.

3rd Editorial Decision

06 October 2016

Thank you for the submission of your revised manuscript to EMBO Molecular Medicine.

Based on your rebuttal concerning Reviewer 2's remaining doubts and after internal discussion, I have decided to proceed with an editorial decision. I am pleased to inform you that we will be able to accept your manuscript pending the following final editorial amendments:

- 1) Every published paper now includes a 'Synopsis' to further enhance discoverability. Synopses are

displayed on the journal webpage and are freely accessible to all readers. They include a short standfirst as well as 2-5 one sentence bullet points that summarise the paper. Please provide the synopsis including the short list of bullet points that summarise the key NEW findings. The bullet points should be designed to be complementary to the abstract - i.e. not repeat the same text. We encourage inclusion of key acronyms and quantitative information. Please use the passive voice. Please attach this information in a separate file or send them by email, we will incorporate it accordingly. You are also welcome to suggest a striking image or visual abstract to illustrate your article. If you do please provide a jpeg file 550 px-wide x 400-px high.

2) We encourage the publication of source data, particularly for electrophoretic gels and blots, with the aim of making primary data more accessible and transparent to the reader. Would you be willing to provide a PDF file per figure that contains the original, uncropped and unprocessed scans of all or at least the key gels used in the manuscript? The PDF files should be labeled with the appropriate figure/panel number, and should have molecular weight markers; further annotation may be useful but is not essential. The PDF files will be published online with the article as supplementary "Source Data" files. If you have any questions regarding this just contact me.

Please submit your revised manuscript within two weeks. I look forward to seeing a revised form of your manuscript as soon as possible.

3rd Revision - authors' response

24 November 2016

Authors made requested editorial changes.

Corresponding Author Name: Deborah Stroka

Manuscript Number: EMM-2015-06089-V2